# Exploring the Robustness of Language Models for Tabular Question Answering via Attention Analysis

**Kushal Raj Bhandari**                                    *bhandk@rpi.edu*
*Department of Computer Science*
*Rensselaer Polytechnic Institute*

**Sixue Xing**                                             *xings@rpi.edu*
*Department of Computer Science*
*Rensselaer Polytechnic Institute*

**Soham Dan**                                    *sohamdan@microsoft.com*
*Microsoft*

**Jianxi Gao**                                             *gaoj8@rpi.edu*
*Department of Computer Science*
*Rensselaer Polytechnic Institute*

**Reviewed on OpenReview:** https://openreview.net/forum?id=PYHIDN9Wuq

## Abstract

Large Language Models (LLMs), already shown to ace various unstructured text comprehension tasks, have also remarkably been shown to tackle table (structured) comprehension tasks without specific training. Building on earlier studies of LLMs for tabular tasks, we probe how in-context learning (ICL), model scale, instruction tuning, and domain bias affect Tabular QA (TQA) robustness by testing LLMs, under diverse augmentations and perturbations, on diverse domains: Wikipedia-based **WTQ**, financial **TAT-QA**, and scientific **SCITAB**. Although instruction tuning and larger, newer LLMs deliver stronger, more robust TQA performance, data contamination and reliability issues, especially on **WTQ**, remain unresolved. Through an in-depth attention analysis, we reveal a strong correlation between perturbation-induced shifts in attention dispersion and the drops in performance, with sensitivity peaking in the model's middle layers. We highlight the need for improved interpretable methodologies to develop more reliable LLMs for table comprehension. Through an in-depth attention analysis, we reveal a strong correlation between perturbation-induced shifts in attention dispersion and performance drops, with sensitivity peaking in the model's middle layers. Based on these findings, we argue for the development of structure-aware self-attention mechanisms and domain-adaptive processing techniques to improve the transparency, generalization, and real-world reliability of LLMs on tabular data.

## 1 Introduction

LLMs, despite being primarily trained on unstructured text, have demonstrated notable capabilities in structured data tasks, such as Tabular Question Answering (TQA). TQA requires models to interpret data presented in tables, demanding strong structural reasoning. TQA specifically challenges models to discern relationships and hierarchies implicit within tabular data, making it an ideal benchmark for structural reasoning capabilities. Assessing how LLMs navigate structured comprehension challenges can provide valuable insights into their robustness and reasoning capabilities (Borisov et al., 2023; Fang et al., 2024). Tabular QA offers clean, controllable structural perturbations (e.g., row/column swaps, transposes) that directly target schema alignment and spatial reasoning while holding topical content constant. In contrast,

long-context QA evaluations introduce confounds from retrieval, chunking, and position bias that make it harder to attribute attention-pattern changes to structure alone. Our focus on TQA thus isolates structure-sensitive behavior in a way that complements long-context frameworks such as NoLiMa (Modarressi et al., 2025).

Recent studies emphasize the importance of robustness evaluations in understanding LLM behavior on structured tasks (Zhou et al., 2024). Specifically, perturbations in tabular structure and content significantly impact model performance, highlighting vulnerabilities that limit the practical reliability of models (Wang et al., 2022; Zhao et al., 2023). Furthermore, Liu et al. (2023) argues that LLMs inherently struggle with structural manipulations, advocating for an integrated approach combining symbolic reasoning to enhance model robustness.

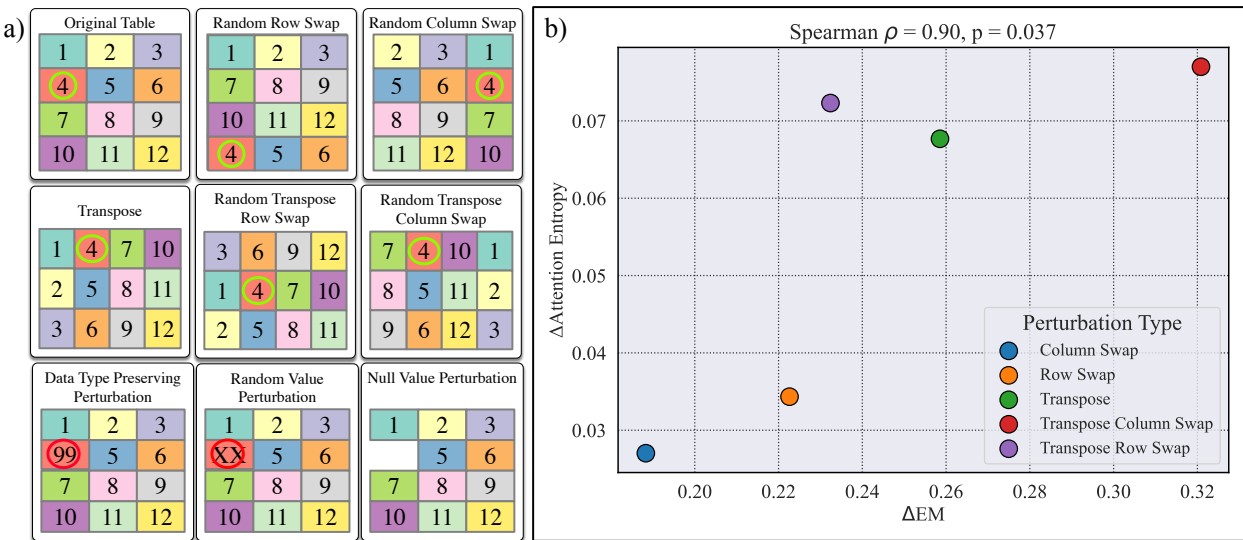

Figure 1: (**a**) An example of the different possible table augmentation methods.(**b**) Scatter plot between change in attention entropy and EM score across perturbation types for `Llama-8B-Instruct` on **WTQ**. The average attention entropy change against EM scores across perturbation types for Llama-8B-Instruct on WTQ, averaged over all attention heads and data points.

Although these studies identify vulnerabilities and suggest potential improvements, there remains limited understanding of how internal model mechanisms respond to perturbations in structured data. Attention mechanisms form the core of transformer-based LLMs, governing how models distribute focus across input elements during processing (Clark et al., 2019). Prior work analyzing attention patterns in natural language tasks revealed that specific layers and attention heads critically influence model performance and robustness (Zhao et al., 2024; Barbero et al., 2025). These attention heads often serve specialized functions such as syntactic parsing or semantic alignment, highlighting the complexity of internal transformer mechanisms. However, detailed attention-level analyses for structured tasks, such as TQA, remain scarce.

To address this gap, we systematically investigate LLM robustness for TQA tasks across multiple dimensions, including *in-context learning*, *model scale*, *instruction tuning*, *domain biases*, *value perturbations*, and *model size*. Our experiments evaluate different perturbation types, highlighting how value-based alterations influence reasoning fidelity and faithfulness (Table 1). We also compare performance across diverse datasets to assess domain biases and generalizability (detailed in Appendix A). Because robustness can be confounded by context length, we explicitly control for table size by restricting all inputs to < 150 cells (Appendix D). We also group-report performance and attention statistics across table-size bins to probe scaling effects in a controlled manner (Appendix F; Fig. 13). Together, these controls allow us to clarify differences primarily to structural understanding rather than to generic long-context limitations (Modarressi et al., 2025).

Extending beyond surface-level performance metrics, we quantify changes in attention entropy across different attention heads and layers under various perturbations, analyzing how these changes correlate

with performance degradation. Attention entropy effectively quantifies the dispersion in a model's attention distribution, providing a nuanced metric for understanding internal decision-making processes. With this, we aim to determine which attention heads are most sensitive to perturbations, leading to more substantial changes, and thus more distinct performance degradation.

Our study leverages diverse datasets: Wikipedia-based **WTQ** (Pasupat & Liang, 2015), financial report-based **TAT-QA** (Zhu et al., 2021), and scientific claims-based **SCITAB** (Lu et al., 2023). Each of these datasets provides a unique context and complexity, allowing us to assess the generalizability and domain-specific sensitivities of LLMs robustly. Through these diverse domains, our findings give insights into the sensitivity and reliability of LLMs on TQA, highlighting the broader challenge of understanding how LLMs reason over structured

## 2 Various Perturbation Categories

Each perturbation is designed to manipulate the table structure or content while preserving the inherent relational meaning of the table, thereby measuring robustness to table comprehension, as illustrated in Fig. 1.

### 2.1 Structural Perturbation (SP):

SP involves rearranging the columns and rows of the table to generate new examples. These perturbations simulate realistic scenarios where data presentation varies significantly, testing the model's ability to comprehend tabular structure. This ensures flexibility in understanding tabular data without distorting the semantics of the table. SP involves column swap, row swap, transpose, transpose column swap, and transpose row swap. These provide diverse perspectives on table comprehension, allowing for a thorough evaluation of the LLMs' ability to handle structural variations.

### 2.2 Value Perturbation (ValP):

ValP focuses on modifying the actual data values within tables, ensuring that the model accurately reflects the data. Value perturbations specifically challenge the semantic fidelity of the model by altering critical data points that directly affect the answer. We explore these types of **ValP**:

> **Data Type Preserving Perturbation (DVP):** DVP involves altering the answer to the question and, respectively, the cell values within the table while maintaining their original data types. For instance, in Fig 9, given a question, "What was the first venue for the Asian games?", we modify the correct answer, "Bangkok, Thailand", to "Beijing". These *counterfactual* entities test the faithfulness of the LLM to the tabular data. [1]. We utilize an automated counterfactual answer generation method that prompts GPT-3.5, ensuring the type correctness of the altered answer. Using a large language model, such fake answer generation makes it possible to generate fake answers that adhere to the data type and make the table semantically correct. Examples of prompts and details of DVP dataset generation are present in the Appendix B and C.

> **Random Value Perturbation (RVP):** RVP(an example is shown in Fig. 10) relaxes DVP where instead of a counterfactual entity, we have a fixed string, e.g., "r@nD0m v@1u3". Performance on this perturbation correlates with whether the injection of random data into the table affects the accuracy. The comparison of random and data type-preserving perturbation also highlights whether models are influenced by the injection of abstract values for table comprehension.

> **Null Value Perturbation (NVP):** NVP removes the correct answer from the table completely. Evaluating the performance on NVP highlights the influence of Wikipedia content on solving the **WTQ** table question-answering task. LLMs that struggle more with the null value perturbation are likely to show consistent performance on TQA across different tabular datasets.

---

[1]We filter out the subset of data points where the table does not contain the answer, e.g., "How many people stayed at least 3 years in office?"

**No Table (NT):** To understand the extent of bias in **WTQ**, we evaluate LLMs on the no-table baseline. This approach further emphasizes the reliance of LLMs on Wikipedia content. By analyzing the performance of LLMs in the absence of the table, we can better understand the extent of dependence on the particular tabular data. This method helps to reveal the intrinsic capabilities of LLMs for TQA and their generalizability across different tabular datasets.

To disentangle background memorization from table use, we employ four value-centric probes. NT and NVP bound performance when factual recall is necessary but local evidence is absent; DVP and RVP stress reliance on value-level fidelity when structure is intact. While a full label-randomization protocol would further isolate memorization effects, it introduces nontrivial semantic-consistency and schema-integrity issues in multi-column tables; we therefore leave a systematic randomization study to future work and use NT/NVP/DVP/RVP as practical, controlled approximations that foreground value-level dependency (related analyses in (Sakai et al., 2024)).

Collectively, these structural and value-based perturbations enable a comprehensive analysis of the model's ability to reason over table structure and content. By introducing such constrained and adversarial transformations, we can more precisely isolate the underlying factors that drive the performance in TQA.

## 3 Evaluation Metrics

Given the definition of different perturbations, we employ these three metrics to evaluate model performance under various perturbations in TQA tasks.

**Exact Match Accuracy (EM)**: This metric calculates the proportion of instances for which the predicted answer exactly matches the ground truth answer. Formally, if $N$ is the total number of instances, and $\mathrm{correct}(i)$ is an indicator function that is 1 if the prediction for instance $i$ is correct and 0 otherwise, then:

$$\mathrm{EM} = \frac{\sum_{i=1}^{N} \mathrm{correct}(i)}{N}.$$

**F1 Measure**: The F1 measure is a harmonic mean of precision and recall, commonly used to evaluate the similarity between two text sequences, such as predicted and ground truth answers. Given two strings, each is tokenized into sets of words, and the overlap between them is used to compute:

$$\mathrm{Precision} = \frac{\text{Number of overlapping tokens}}{\text{Number of tokens in predicted string}}, \quad \mathrm{Recall} = \frac{\text{Number of overlapping tokens}}{\text{Number of tokens in ground truth string}}.$$

The F1 score is then computed as:

$$\mathrm{F1} = 2 \cdot \frac{\mathrm{Precision} \cdot \mathrm{Recall}}{\mathrm{Precision} + \mathrm{Recall}}.$$

A higher F1 score indicates greater similarity between the predicted and ground truth strings, while a lower F1 score reflects greater divergence.

**Exact Match Difference (Emd)**(Zhao et al., 2023; Zhou et al., 2024): Let $\mathrm{EM}_{\mathrm{orig}}$ be the EM on the original (unperturbed) dataset, and $\mathrm{EM}_{\mathrm{perturbed}}$ be the EM after applying a perturbation. The Emd quantifies the change in EM due to perturbations:

$$\mathrm{Emd} = \mathrm{EM}_{\mathrm{perturbed}} - \mathrm{EM}_{\mathrm{orig}}.$$

Negative values indicate performance degradation, while values close to zero imply robustness against perturbations.

**Variation Percentage (VP)**(Yang et al., 2022; Zhou et al., 2024): This metric measures the extent to which predictions change after applying perturbations. Given that, *C2W* is the count of correct

Table 1: The average EM, F1, VP, and Emd of small and large LLMs on Value Perturbations(**ValP**). The classification of large and small models is defined in Table 3.

| ValP | Large Model | | | | Small Model | | | |
|---|---|---|---|---|---|---|---|---|
| **Operation** | EM | F1 | VP | Emd | EM | F1 | VP | Emd |
| **WTQ** | | | | | | | | |
| **Original** | 0.37 | 0.45 | - | - | 0.26 | 0.34 | - | - |
| **NT** | 0.05 | 0.06 | 0.37 | -0.32 | 0.02 | 0.03 | 0.26 | -0.24 |
| **DVP** | 0.22 | 0.29 | 0.40 | -0.14 | 0.14 | 0.20 | 0.31 | -0.12 |
| **RVP** | 0.15 | 0.20 | 0.39 | -0.22 | 0.10 | 0.14 | 0.30 | -0.16 |
| **NVP** | 0.06 | 0.09 | 0.37 | -0.30 | 0.03 | 0.05 | 0.27 | -0.23 |
| **TAT-QA** | | | | | | | | |
| **Original** | 0.34 | 0.43 | - | - | 0.21 | 0.27 | - | - |
| **NT** | 0.01 | 0.04 | 0.34 | -0.33 | 0.00 | 0.02 | 0.20 | -0.20 |
| **SCITAB** | | | | | | | | |
| **Original** | 0.113 | 0.116 | - | - | 0.135 | 0.137 | - | - |
| **NT** | 0.000 | 0.00 | 0.120 | -0.120 | 0.000 | 0.00 | 0.132 | -0.132 |

before perturbation and wrong after, and *W2C* is the count of wrong before perturbation and correct after. Given $N$ as the total number of instances, the variation percentage is:

$$\text{VP} = \frac{C2W + W2C}{N}.$$

A higher VP indicates greater sensitivity of predictions to perturbations, while a lower VP signifies more stable predictions.

In addition to performance-based metrics, we also examine internal model behavior through the analysis of attention patterns.

**Attention Entropy**: We analyze attention entropy to capture the dispersion of attention within different attention heads. Entropy serves as a measure for structural awareness, where high entropy indicates more evenly distributed attention, while low entropy reflects concentrated focus on a few tokens.

$$H_i = -\sum_j \mathbf{A}_{ij} \log(\mathbf{A}_{ij})$$

where $\mathbf{A}_{ij}$ is the attention weight to the $j$-th token.

# 4 Evaluation Performance

## 4.1 Effects of ICL examples on TQA

*Does instruction prompting assist LLM for better table comprehension for question-answering tasks?* The heat maps in Figure 2 illustrate the performance of various LLMs, demonstrating that models fine-tuned for instructions or conversation exhibit improved performance. For instance, the `Llama3-70B-Instruct` model significantly outperforms its original version across all table transformations, indicating that instruction-based fine-tuning enhances the model's ability to handle complex reasoning tasks. Similarly, conversation-focused fine-tuning also yields better scores, albeit with a less pronounced improvement compared to instruction-focused tuning. This suggests that fine-tuning models on specific tasks like following instructions or conversing effectively enhances their capability to interpret and manipulate tabular data, making such approaches valuable for improving performance in structured data tasks. For Llama2 models, the gains from instruction-tuning are present but less pronounced compared to Llama3, especially on **TAT-QA** where the performance gap between base and instruction-tuned variants is narrower. Qwen2.5 models exhibit a similar improvement trend to Llama3, but their relative advantage is most visible on **SCITAB**, where instruction-tuned variants close a larger fraction of the base-to-top model gap. Qwen3 shows decent performance on **WTQ**, and also similar performance with the **Base** variant, suggesting that its base configuration already incorporates stronger

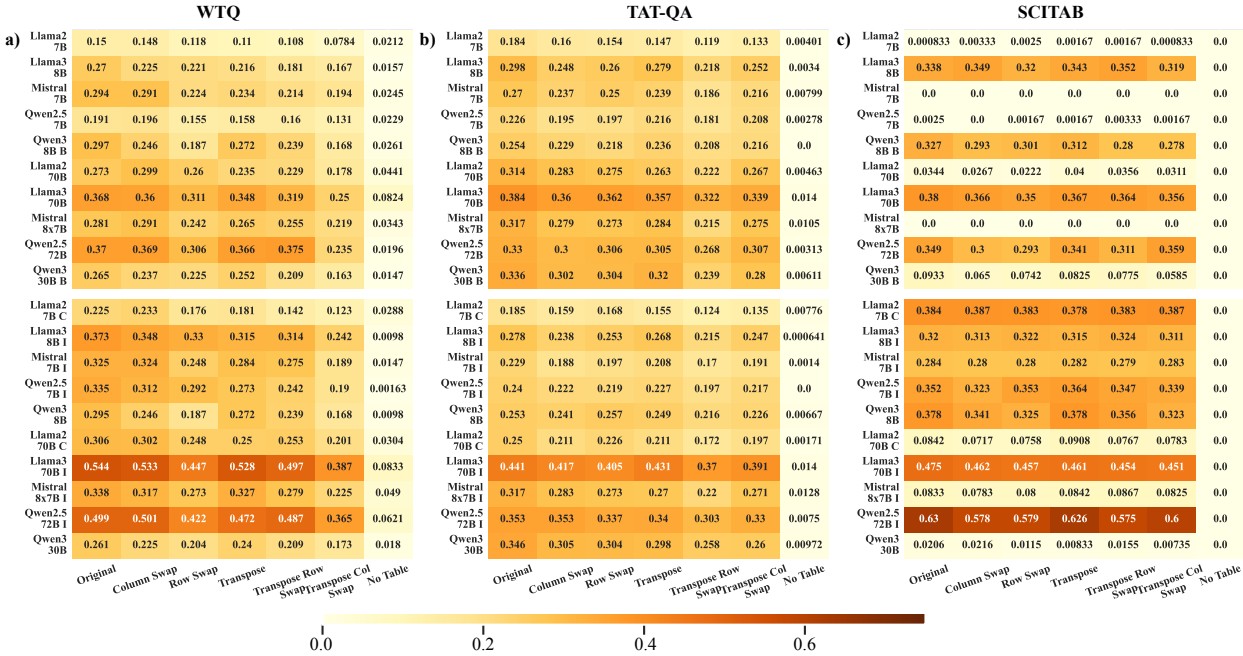

Figure 2: The average EM scores of the original/base(denoted by **B**) models and conversation(denoted by **C**)/instruction(denoted by **I**) models across various table augmentation techniques for **WTQ**, **TAT-QA** and **SCITAB** dataset with three fewshot examples.More individual results with additional metric are included in Appendix G.

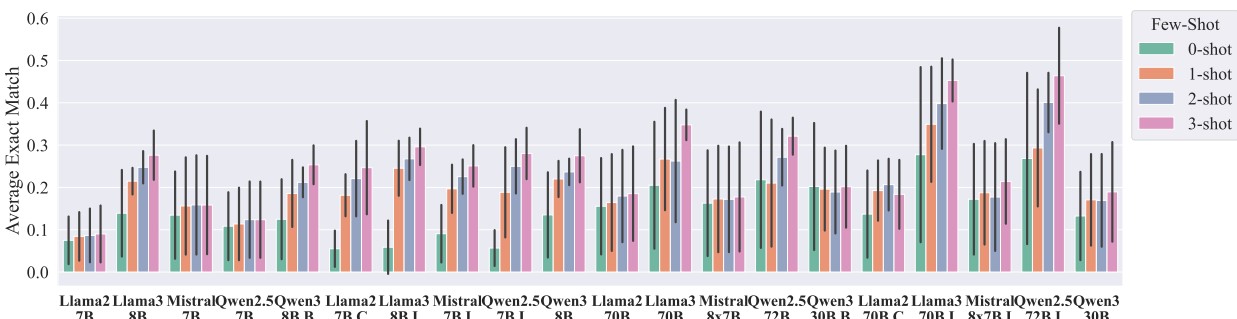

Figure 3: Comparison of average EM scores across models under varying few-shot settings (0-shot, 1-shot, 2-shot, and 3-shot). Here, **B** denotes the Base variant of the model, **C** denotes the Chat variant, and **I** denotes the Instruction variant.

reasoning priors for table tasks. It also distinctly indicates that LLMs that have undergone instruction or conversation-based fine-tuning outperform their base counterparts in TQA tasks for **SCITAB** dataset. Although **SCITAB** is inherently a classification task presented in a TQA format, using few-shot prompting provides valuable context, ultimately leading to more accurate and relevant responses.

## 4.2 Effects of the Model Type on TQA

*Do newer models have better TQA abilities?* Figure 4 shows that Llama3 models generally outperform the Llama2 and Mistral models across different configurations, indicating that newer architectures like Llama3 are more effective at table reasoning tasks. The 70B versions of these models generally perform better than their 7B variants, indicating that larger model sizes enhance reasoning capabilities. Overall, the advancements in model architecture and increased model size significantly contribute to better TQA abilities. While Llama3

models outperform their predecessors, the margin over Llama2 is largest on **WTQ** and narrows on **TAT-QA** and **SCITAB**. Notably, Qwen2.5 models, especially the larger variants, perform competitively with Llama3 on **TAT-QA**, often surpassing Mistral. For smaller variant models, Qwen3 shows comparable performance to Qwen2.5 model; however, the larger Qwen2.5 model retains an absolute performance edge in most cases. Larger models (e.g., Llama3-70B, Mixtral-8x7B, Qwen2.5-72B) generally show higher performance than

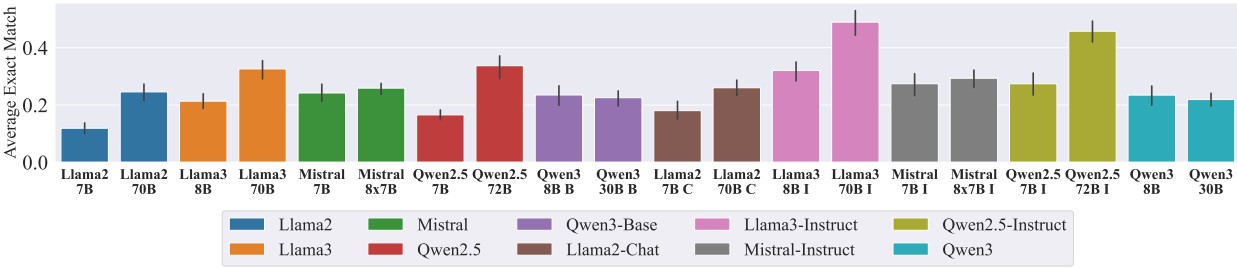

Figure 4: The average EM for different models across the **WTQ** dataset with 3-fewshot settings over all structural perturbation. The models are categorized by their types (Llama2, Llama3, Mistral, Qwen2.5 and Qwen3) and separated by their sizes (7B, 70B, etc.)

smaller models (e.g., Llama3-8B, Mistral-7B, Qwen2.5-8B), as seen in both the bar plot (Figure4) and heat maps (Figure2). For instance, Llama3-70B and Mixtral-8x7B have higher EM scores than Llama3-8B and Mistral-7B. However, we do see that average EM between the Qwen3 variant of small and large variants is not significantly different. This could be the result of an architectural difference between a dense model and MOE model. Although large, the active parameters in the 30B parameter model are only 3B parameter model. This suggests that model size has a significant impact on TQA performance.

*How do performances vary with value perturbations?* [2] Table 1 shows the performance of various LLMs, as defined on Table 3, when subjected to different value-based perturbations on **WTQ** using different evaluation metrics: Exact Match(EM), Variation percentage (VP), and Exact Match Difference (EMD). We observe that LLMs experience a decline in EM scores across various operations compared to the original setup. The performance with RVP results in a significant performance drop, more so than DVP. This suggests a sharp decrease in the model's ability to process and comprehend tables when the insertion of arbitrary, non-contextual values compromises the table comprehension ability of LLMs. Conversely, the DVP result indicates that while the model struggles with content that deviates from the original data structure, maintaining data type consistency helps. VP increases significantly across all perturbations, indicating considerable changes in predictions due to the perturbations. EMD consistently shows negative values, with the most substantial performance drops occurring in the NT and NVP scenarios. The observed discrepancies in performance, particularly pronounced in DVP and RVP, underline a fundamental challenge: these models do not consistently apply their tabular comprehension capabilities when faced with perturbed tables.

## 4.3 Effects of Domain Specificity on TQA

*How biased are LLMs towards Wiki-tables?* Table 1 shows that when no table is provided, LLMs show a notable performance decline, emphasizing their dependence on tabular data to generate correct answers. Interestingly, the models still manage to answer about ≈5% of queries correctly, indicating a potential bias in Wiki-data. In the NVP scenario, where table values relevant to queries are nullified, there is a significant drop in performance, yet less severe compared to the complete absence of a table. This suggests that the models are biased by the contextual structured format cue even in the absence of relevant data.

*How do out-of-box LLMs perform on specialized domains: TAT-QA and SCITAB?* As shown in Table 1, LLMs exhibit moderate performance across various table augmentations on the **TAT-QA** dataset and **SCITAB** dataset. For **TAT-QA**, large models consistently outperform smaller models, underscoring their superior TQA abilities, but for **SCITAB**, the small models have better performance in comparison to larger

---

[2]More details are in Appendices F

models. The overall inconsistent scores still denote significant challenges inherent in niche domains, but instruction-tuned models are better here. From Table 1, the comparison among **WTQ**, **TAT-QA**, and **SCITAB** datasets reveals an interesting *accuracy-robustness tradeoff*: while models with higher EM on **WTQ** suffer larger EMD and higher VP, indicating higher sensitivity to perturbations, their relatively smaller EMD and lower VP on both **TAT-QA** and **SCITAB** reflect greater robustness, despite **SCITAB**'s overall lower baseline EM. Also notable is the sharp contrast in the NT performance, where on **TAT-QA** and **SCITAB** datasets, models have approximately 0% accuracy, a contrast from the **WTQ** performance. This suggests that performance on **WTQ** might be inflated due to biases favoring familiarity with Wikipedia, compared to niche domains like **TAT-QA** and **SCITAB**. This highlights the need for better benchmark design for tabular understanding. We also observe trends consistent with prior work: performance improves with few-shot prompting, instruction-tuned models generally outperform their base counterparts, and larger models tend to exhibit stronger TQA (Wei et al., 2022; Fang et al., 2024).

## 5 Attention Analysis

Understanding how structural perturbations in tabular inputs affect a model is critical for assessing robustness and diagnosing potential failure modes in TQA. As attention mechanisms regulate how language models allocate focus across table elements, examining their sensitivity to perturbations provides insights into representational stability. Here, we analyze this sensitivity by quantifying how perturbation-induced changes in attention dispersion correlate with performance degradation.

We utilize *attention entropy* as a quantitative measure of how attention distributions respond to structural perturbations. Attention entropy reflects the dispersion of attention weights across input tokens, serving as an indicator of how focused or diffuse the model's attention becomes under disruption. In the context of tabular data, where semantic understanding hinges on precise structural alignment, operations like row and column swaps inherently distort positional cues. These distortions lead to measurable shifts in entropy, making it a particularly effective diagnostic for attention misalignment. By tracking how entropy changes in response to perturbations, we capture the extent to which structural instability propagates through the attention mechanism. This perspective adds on prior work showing the role of entropy in capturing context encoding diversity Zhang et al. (2025), stabilizing attention dynamics during training Zhai et al. (2023), and improving generalization under limited data conditions Araabi et al. (2024), and extends these insights to structured, table-based inputs.

### 5.1 Effect of Perturbations on Attention Maps

We examine how varying severities of structural perturbations influence attention weights using the `Llama3-8B-Instruct` model on the **WTQ** dataset. For each attention head across all layers, we measure the change in attention entropy between the original and perturbed table inputs. In parallel, we compute the corresponding change in Exact Match (EM) scores, capturing the performance impact of each perturbation.

*Attention entropy* captures the distribution uniformity of attention across tokens; higher entropy indicates diffuse attention, while lower entropy signifies focused attention. Thus, significant entropy changes reflect considerable shifts in the model's internal attentional focus due to perturbations.

Our analysis (Figure 1) reveals a significant positive correlation (Spearman $\rho = 0.90$, $p = 0.037$) between changes in attention entropy and EM degradation across perturbation types. This shows that more severe perturbations significantly disrupt the model's attention distribution, consequently diminishing its ability for TQA.

These findings highlight the sensitivity of attention mechanisms to structural integrity within tabular inputs. Perturbations affecting relational semantics induce greater attention dispersion and misalignment, directly impairing model accuracy. Thus, attention dispersion is a crucial indicator linking input perturbations with performance.

## 5.2 Analysis of Individual Attention Head Sensitivity to Perturbations

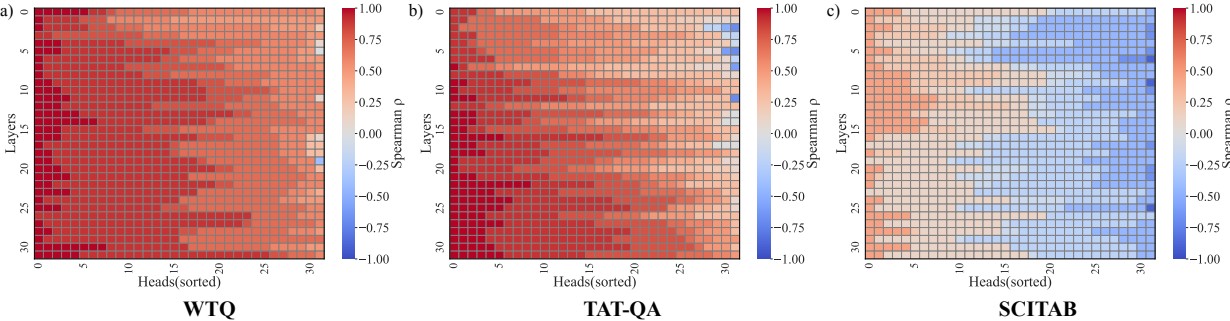

Figure 5: Heatmap showing Spearman correlation between changes in attention entropy and EM difference across all attention heads and layers in the `Llama3-8B-Instruct` model on the **a) WTQ**, **b) TAT-QA**, and **c) SCITAB** dataset

While averaging attention metrics across heads provides a general understanding of model behavior, prior research emphasizes that individual attention heads and specific layers, particularly the middle layers, significantly contribute to encoding task-relevant information (Barbero et al., 2025; Zhao et al., 2024). To investigate this at a finer granularity, we perform a layer-wise analysis of how perturbation-induced changes in attention entropy correlate with EM score degradation.

Specifically, we use the `Llama3-8B-Instruct` model on the **WTQ** dataset to compute the Spearman correlation between entropy changes (original vs. perturbed inputs) and corresponding EM differences for each attention head across all layers. Figure 5(**a**) demonstrates that correlations peak predominantly in the middle layers and remain consistently elevated through these central layers. This indicates that perturbation-induced shifts in attention distribution within middle layers are strongly predictive of performance degradation. The sharp correlation peak in **WTQ** suggests that the model relies heavily on mid-layer representations to align tabular structures with natural language queries, particularly when interpreting entity references and table schema. Distinct, though narrower, spikes at the input layer (0) and the output layers (30–31) further reveal that both the earliest token-encoding stage and the final representational consolidation phase are also vulnerable to structural perturbations.

Figure 5(**b**) and 5(**c**) extend the analysis to **TAT-QA** and **SCITAB**, respectively. The results reveal distinct patterns tied to domain characteristics. For **TAT-QA**, correlations are elevated not only in the middle layers but also extend into the upper layers of the model. In contrast, **SCITAB** exhibits a more sharply localized pattern, with peak correlations tightly concentrated within the middle layers. Although we observe a high correlation in these middle layers, the contrastive negative correlations are primarily due to the significantly poor EM performance on the **SCITAB** dataset. These findings reinforce the centrality of middle-layer attention mechanisms in tabular reasoning tasks, while also highlighting domain-specific variations in the vertical distribution of sensitivity. Additional experiments are included within the Appendix H for other models(`Llama3-8B`, `Mistral-7B`, and `Mistral-7B-Instruct`).

## 5.3 Correlation within Individual Attention

We find that structural perturbations in tabular inputs affect the internal attention dynamics of LLMs, specifically the `Llama3-8B-Instruct` model on **WTQ**, and that these dynamics are differentially expressed across the attention heads and layers. Figure 6(**a**) presents an unsorted Spearman correlation heatmap between per-head changes in attention entropy and the corresponding EM score differences, revealing substantial heterogeneity across all 32 layers and 32 heads. The heatmap captures a wide range of correlation strengths, from near-zero to values approaching $\rho = 0.9$, illustrating that not all attention mechanisms contribute equally to robustness. In particular, the concentration of high-correlation values in lower layers suggests that early-stage attention heads may play a foundational role in establishing reliable structural interpretations.

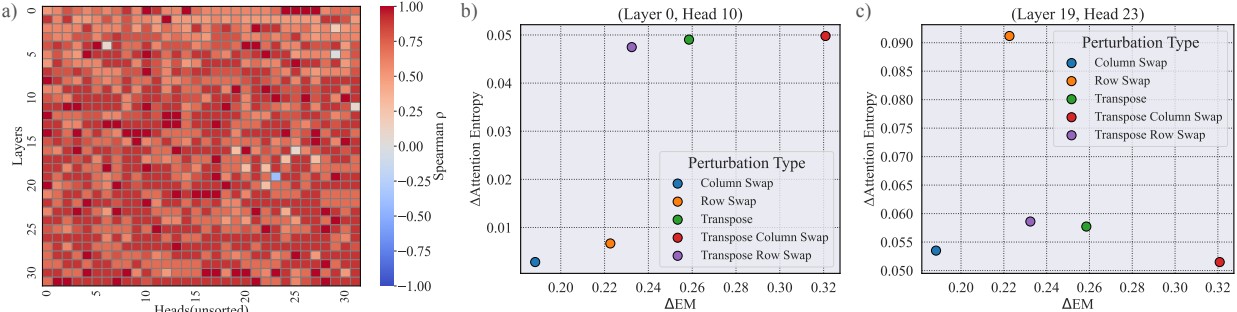

Figure 6: **(a)** Heatmap showing Spearman correlation between changes in attention entropy and EM difference across all attention heads and layers in the `Llama-8B-Instruct` model on the **WTQ** dataset, unsorted to depict actual correlation over each attention head. **(b-c)** Scatter plot showing the correlation between changes in attention entropy and EM scores across five structural perturbations for the **WTQ** dataset. **(b)** The plot highlights a strong positive correlation for Layer 0, Head 10, indicating high sensitivity of this head to structural perturbations. **(c)** In contrast, Layer 19, Head 23 shows the least correlation. Note that each point in plots **b** and **c** averages only across data points for each specific head-layer pair.

To unpack this variability, Figures 6**(b)** and 6**(c)** zoom in on two extreme heads identified from the heatmap. Layer 0, Head 10 (Figure 6**(b)**) exhibits a strong positive relationship: higher shifts in attention entropy due to structural perturbations reliably predict larger drops in EM scores. This implies that this head is critical for encoding spatial consistency or row-column alignment in tabular inputs, and disruptions to this alignment directly degrade model accuracy. The correlation is visually evident as a tight clustering of points around a positively sloped trend line across all five structural perturbation types (Row Swap, Column Swap, Transpose, Transpose Row Swap, and Transpose Column Swap).

In contrast, Layer 19, Head 23 (Figure 6**(c)**) demonstrates minimal correlation between entropy change and EM variation. This indicates a form of functional redundancy or robustness in this head's role; it either performs a task unrelated to structural parsing or maintains stable attention regardless of structural distortions. This wide range in sensitivity reinforces the notion that attention heads are functionally diverse and that only a subset contributes significantly to robustness under tabular perturbation.

Altogether, these findings suggest the possibility of identifying and selectively reinforcing robustness-critical attention heads through architectural tuning or fine-tuning objectives. By mapping correlation strengths across the entire model, we can isolate those mechanisms most affected by structural shifts and potentially develop strategies, such as attention regularization or selective re-weighting, that mitigate their susceptibility to disruption. This also opens avenues for pruning or interpretability studies focused on attention heads with minimal impact, offering insights into model compression or simplification without significant performance loss.

## 5.4 Cross-Model and Dataset Analysis of Perturbation Sensitivity

To evaluate the generality of our findings beyond the `Llama3-8B-Instruct` model on WTQ, we extend our correlation analysis between attention entropy change and EM performance degradation across a diverse set of model-dataset combinations. These include all the small models defined in 3, for which each models are tested on the **WTQ**, **TAT-QA**, and **SCITAB** datasets. For each configuration, we compute the Spearman correlation between the perturbation-induced changes in attention entropy and the corresponding drop in EM scores.

As shown in Figure 7, the positive correlation trend persists across nearly all settings. In particular, `Instruct` and `chat` variants of the models consistently show the strongest correlation values across datasets, reinforcing their heightened sensitivity to attention dispersion caused by structural perturbations. Notably, even non-instruct variants like `Mistral-7B` exhibit moderate to strong correlations, although the magnitude tends to be

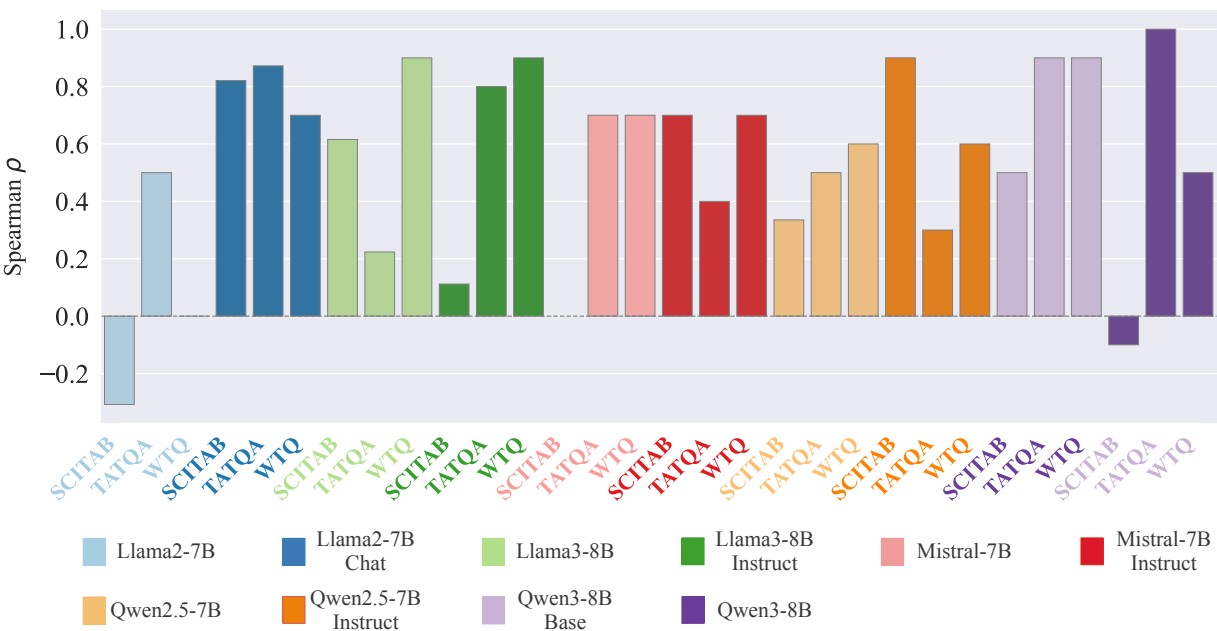

Figure 7: Spearman correlation between perturbation-induced attention-entropy change and EM degradation across 30 model–dataset pairs.

lower and more variable. We also find that the **SCITAB** dataset exhibits the greatest variation in correlation values, specifically due to the particularly poor performance of the `Llama2-7B` model on this dataset, where the baseline EM with the original table is already low enough to substantially affect the correlation analysis.

These consistent patterns across models and datasets confirm the robustness of our main claim: the severity of perturbation that induces greater shifts in attention entropy is reliably associated with a decline in model performance. This suggests that the change in attention entropy serves as a broadly applicable proxy for evaluating robustness in table-based QA models.

Moreover, this analysis demonstrates that the observed dynamics are not specific to any single dataset or model architecture. Instead, they reflect a more general representational vulnerability within current attention-based architectures when handling perturbed structured inputs.

## 6 Conclusion

Our study provides a comprehensive and robust analysis of LLMs under various perturbations for TQA. While larger, instruction-tuned models show improved performance, they remain highly sensitive to structural and value-based disruption. These disruptions notably manifest as perturbations such as random row swaps, column swaps, and transpositions, highlighting vulnerabilities in their structural reasoning capabilities. We also uncover domain biases, where models perform well on **WTQ** without tables but struggle on more specialized datasets, such as **TAT-QA** and **SCITAB**. Specifically, the performance on **WTQ** even in the absence of tables indicates reliance on memorized textual patterns from pretraining, rather than genuine tabular reasoning. In contrast, specialized domains such as financial and scientific tables pose more significant challenges due to their complexity and unique domain specificity.

We demonstrate that shifts in attention entropy, particularly in middle layers, are correlated with performance degradation. This observation was supported by detailed attention-level analyses, which revealed that attention heads in middle layers are particularly critical in encoding structural information, and their instability directly contributes to errors in comprehension. Furthermore, layer-wise attention analysis reveals

that certain attention heads exhibit greater sensitivity, suggesting that targeted improvements in these areas could enhance robustness.

Our findings underline the critical need for improving both the interpretability and robustness of LLMs in TQA. Building on these insights, future research should focus on developing fine-grained attention-interpretability methods and domain-adaptive processing strategies to enhance transparency, cross-domain generalization, and reliability in practical applications.

# 7 Limitation

Although we provide extensive evaluation of LLMs on **WTQ**, **TAT-QA**, and **SCITAB** datasets, it is possible to include a broader range of datasets for a more comprehensive comparison that would highlight the generalizability of our method for both domain-specific datasets and Wikipedia-based datasets. While we anticipate that similar performance could be achieved with other tasks, such as table summarization, future work should include extensive analysis across various tasks and datasets to validate the assumption. It would also be insightful to expand the analysis to make a comparison with long-context QA. Moreover, our study did not involve any detailed analysis on structurally aware or fine-tuned models for tabular datasets. It is plausible that fine-tuning and structurally enhanced models could significantly impact the performance of different models. Additionally, our evaluation relied on exact match accuracy for assessing the text generation model's performance. This metric, while useful, limits the scope of evaluation for the question answering task. Future studies should employ more nuanced evaluation metrics to better assess the robustness of the models in TQA tasks. Moreover, we only conduct a case study of attention analysis with small models because of the computation cost.

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

# A    Dataset

We study three tabular QA datasets chosen for complementary domains and reasoning demands. **WTQ** (Wikipedia tables) emphasizes broad-coverage knowledge with mixed textual–numeric cells and multi-step, compositional queries over semi-structured schemas. **TAT-QA** (financial reports) combines dense numeric content with accompanying textual descriptions, stressing arithmetic reasoning and text–table fusion. **SC-ITAB** (scientific tables) features claims supported by measurements and categorical attributes, prioritizing structured logical inference under domain-specific vocabulary. To limit length-induced degradation and focus the analysis on table comprehension, we cap all tables at $< 150$ cells (Appendix D). We further stratify results by table-size bins (Appendix F; Fig. 13) to examine how robustness and attention behavior evolve with increasing structural complexity.

### WTQ dataset

**WTQ** features tables taken from Wikipedia and paired with natural language questions that test a model's ability to reason across semi-structured data. This dataset is especially useful for evaluating two key aspects of question answering: the breadth of domain knowledge and the depth of logical reasoning. Unlike datasets that focus on only one of these, **WTQ** challenges models on both fronts. Because the tables come from a wide range of topics on Wikipedia, **WTQ** introduces semantic variability, which tests a model's ability to generalize beyond surface-level patterns.

### TAT-QA dataset

**TAT-QA** blends structured financial tables with associated unstructured textual explanations, challenging models to align table content with surrounding text. The questions frequently require arithmetic computations (e.g., difference, sum, ratio) and integration of multiple modalities. This dataset simulates real-world information extraction tasks where decision-making depends on both tabular facts and contextual narrative, making it ideal for evaluating text-table fusion capabilities and numerical robustness. Errors in **TAT-QA** tend to reveal how well models can track quantitative facts across modalities.

### SCITAB dataset

**SCITAB** introduces TQA grounded in scientific inquiry, where tables often encode structured results from controlled experiments or observational studies. The dataset features complex relational structures, such as columnar trends, controlled variables, and statistical comparisons. Unlike **WTQ** or **TAT-QA**, **SCITAB** emphasizes scientific reasoning, requiring models to interpret claims, draw evidence-backed conclusions, and perform multi-hop inferences within a structured format. The nuanced scientific language and evidence-based justification make **SCITAB** a strong benchmark for testing logical rigor and attention to detail in LLMs.

# B    Example Prompt

Example of the prompts:

# C    Generating Data Type Preserving Perturbation Dataset

We utilize ChatGPT-3.5 to generate type-preserving counterfactual answers using the prompt illustrated in Fig. 12. The prompt is designed to ensure that any generated answer aligns with the data type of the original answer while deliberately introducing a perturbation. For instance, given a question like "What was the first venue for the Asian games?" with the correct answer "Bangkok, Thailand," the model is prompted to output a different but type-consistent response, such as "Beijing." Similarly, for numerical data, a value like "42" might be replaced with another plausible number, such as "50," ensuring that the altered answer retains the original datatype constraints. This is achieved by explicitly defining the model's role as a generator of "fake" answers that preserve the data type of the original value. To ensure the validity of value perturbations,

```
Based on the information shown in the Table, answer the following Test
Question.

Ensure the final answer format is only 'Final Answer: AnswerName1,
AnswerName2...' form, no other form.
Test:

Table
| Year | Competition         | Venue                  | Position | Notes  |
| 1996 | Olympic Games       | Atlanta, United States | 36th (q) | 5.55 m |
| 1998 | Asian Games         | Bangkok, Thailand      | 8th      | 6.07 m |
| 1999 | World Championships | Seville, Spain         | 23rd (q) | 6.40 m |
| 2000 | Olympic Games       | Sydney, Australia      | 14th (q) | 6.57 m |
| 2001 | World Championships | Edmonton, Canada       | 13th (q) | 6.46 m |
| 2002 | Asian Championships | Colombo, Sri Lanka     | 1st      | 6.61 m |
| 2002 | Asian Games         | Busan, South Korea     | 3rd      | 6.30 m |
| 2003 | World Championships | Paris, France          | 23rd (q) | 6.13 m |
| 2003 | Asian Championships | Manila, Philippines    | 6th      | 6.23 m |
| 2004 | Olympic Games       | Athens, Greece         | 11th     | 6.53 m |

Question: What was the first venue for the Asian Games?

Final Answer: Bangkok, Thailand
```

Figure 8: Example of a prompt with answer for **WTQ** dataset without Few Shot Prompt

```
Based on the information shown in the Table, answer the following Test
Question.

Ensure the final answer format is only 'Final Answer: AnswerName1,
AnswerName2...' form, no other form.
Test:

Table
| Year | Competition         | Venue                  | Position | Notes  |
| 1996 | Olympic Games       | Atlanta, United States | 36th (q) | 5.55 m |
| 1998 | Asian Games         | Beijing                | 8th      | 6.07 m |
| 1999 | World Championships | Seville, Spain         | 23rd (q) | 6.40 m |
| 2000 | Olympic Games       | Sydney, Australia      | 14th (q) | 6.57 m |
| 2001 | World Championships | Edmonton, Canada       | 13th (q) | 6.46 m |
| 2002 | Asian Championships | Colombo, Sri Lanka     | 1st      | 6.61 m |
| 2002 | Asian Games         | Busan, South Korea     | 3rd      | 6.30 m |
| 2003 | World Championships | Paris, France          | 23rd (q) | 6.13 m |
| 2003 | Asian Championships | Manila, Philippines    | 6th      | 6.23 m |
| 2004 | Olympic Games       | Athens, Greece         | 11th     | 6.53 m |

Question: What was the first venue for the Asian Games?

Final Answer: Beijing
```

Figure 9: Example of a prompt with answer for **WTQ** dataset for Data Type Preserving Perturbation. In comparison to Fig. 8, we replace the correct answer(**Bangkok, Thailand**) with a fake answer(**Beijing**).

we select only tables that contain the exact answer to the given question. These constraints guarantee that data type preserving perturbations are applied exclusively to table-question-answer triples where the table explicitly contains the correct answer required to solve the question.

```
Based on the information shown in the Table, answer the following Test
Question.

Ensure the final answer format is only 'Final Answer: AnswerName1,
AnswerName2...' form, no other form.
Test:

Table
| Year | Competition        | Venue                 | Position  | Notes  |
| 1996 | Olympic Games      | Atlanta, United States | 36th (q) | 5.55 m |
| 1998 | Asian Games        | r@nD0m v@1u3          | 8th       | 6.07 m |
| 1999 | World Championships | Seville, Spain        | 23rd (q) | 6.40 m |
| 2000 | Olympic Games      | Sydney, Australia     | 14th (q) | 6.57 m |
| 2001 | World Championships | Edmonton, Canada      | 13th (q) | 6.46 m |
| 2002 | Asian Championships | Colombo, Sri Lanka    | 1st       | 6.61 m |
| 2002 | Asian Games        | Busan, South Korea    | 3rd       | 6.30 m |
| 2003 | World Championships | Paris, France         | 23rd (q) | 6.13 m |
| 2003 | Asian Championships | Manila, Philippines   | 6th       | 6.23 m |
| 2004 | Olympic Games      | Athens, Greece        | 11th      | 6.53 m |

Question: What was the first venue for the Asian Games?

Final Answer: r@nD0m v@1u3
```

Figure 10: Example of a prompt with answer for **WTQ** dataset for Random Value Perturbation. Here, we replace the correct answer(**Bangkok, Thailand**) with an abstract random value (**r@nD0m v@1u3**).

```
Based on the information shown in the Table, answer the following Test
Question.

Ensure the final answer format is only 'Final Answer: AnswerName1,
AnswerName2...' form, no other form.
Test:

Table
| Year | Competition        | Venue                 | Position  | Notes  |
| 1996 | Olympic Games      | Atlanta, United States | 36th (q) | 5.55 m |
| 1998 | Asian Games        |                       | 8th       | 6.07 m |
| 1999 | World Championships | Seville, Spain        | 23rd (q) | 6.40 m |
| 2000 | Olympic Games      | Sydney, Australia     | 14th (q) | 6.57 m |
| 2001 | World Championships | Edmonton, Canada      | 13th (q) | 6.46 m |
| 2002 | Asian Championships | Colombo, Sri Lanka    | 1st       | 6.61 m |
| 2002 | Asian Games        | Busan, South Korea    | 3rd       | 6.30 m |
| 2003 | World Championships | Paris, France         | 23rd (q) | 6.13 m |
| 2003 | Asian Championships | Manila, Philippines   | 6th       | 6.23 m |
| 2004 | Olympic Games      | Athens, Greece        | 11th      | 6.53 m |

Question: What was the first venue for the Asian Games?

Final Answer: Bangkok, Thailand
```

Figure 11: Example of a prompt with answer for **WTQ** dataset for Random Value Perturbation. We remove the correct answer(**Bangkok, Thailand**).

```
Role [System]:
You are a fake answer generator that outputs
fake answer to a given question. You will only
provide a one word answer but match the
datatype.

Role [User]:
Provide a fake answer by matching the datatype,
if it is a number provide a similar number, if
it is a location provide a fake location, and
if it a word then provide a different word.
Given the Question: {Question}
Provide a one word incorrect answer:
```

Figure 12: Prompt designed for generating one-word, datatype-matching fake answers to questions from the **WTQ** dataset.

## D  Evaluation Dataset Size

We select three different datasets for comparison, **WTQ**, **TAT-QA**, and **SCITAB** datasets, with different numbers of evaluation datasets as described in Table 2. Each dataset provides distinct characteristics, **WTQ** is composed of Wikipedia tables, **TAT-QA** centers on financial reports, and **SCITAB** focuses on scientific claims, which together enable a comprehensive and robust assessment. For fair comparison, we limit the number of cell elements ($< 150$) within the table for both datasets. This constraint ensures that models are not disproportionately affected by excessively large tables, which could skew performance due to context window limitations. Moreover, keeping table size bounded allows for more consistent measurement of the effects of perturbations across datasets.

Similarly, for Value Perturbation, some queries relate to the overall structure of the table. Hence, we filter only those tables that contain the answer value for the given query. This filtering ensures semantic alignment between the question and the modified table, avoiding misleading evaluations where no correct answer exists in the perturbed version. In the case of the **WTQ** dataset, for instance, not all table-question pairs are amenable to value alterations, particularly when the table structure is insufficiently informative or lacks direct answer candidates. Altogether, this preprocessing pipeline yields a curated benchmark that is well-suited for analyzing perturbation sensitivity while controlling for confounding factors related to table size and answerability.

| Operation | Number of Pairs |
|---|---|
| **WTQ** dataset | |
| Row Swap | 204 |
| Column Swap | 204 |
| Transpose | 204 |
| Transpose Row Swap | 204 |
| Transpose Column Swap | 204 |
| Data Type Preserving | 141 |
| Random Value | 141 |
| Null Value | 141 |
| No Table | 204 |
| **TAT-QA** dataset | |
| Row Swap | 1668 |
| Column Swap | 1668 |
| Transpose | 1668 |
| Transpose Row Swap | 1668 |
| Transpose Column Swap | 1668 |
| No Table | 1668 |
| **SCITAB** dataset | |
| Row Swap | 1225 |
| Column Swap | 1225 |
| Transpose | 1225 |
| Transpose Row Swap | 1225 |
| Transpose Column Swap | 1225 |
| No Table | 1225 |

Table 2: Size of the Evaluation datasize

## E  Models

We selected recent open-source models that have been extensively studied and analyzed. Table 3 lists all the models we considered with their parameter size and their date of release. We include both base and instruction-tuned variants, allowing us to explore not only the effect of scale but also the impact of task specialization on tabular reasoning. The models span three major families: `Llama-2`, `Llama-3`, `Mistral`, `Qwen2.5` and `Qwen3`, which together represent some of the most widely adopted transformer architectures in the open-source ecosystem. The inclusion of instruction-tuned variants is particularly important, as these models are optimized for following natural language instructions. This ability has been shown to influence performance on tasks that require a structured understanding significantly.

Moreover, by categorizing the models into 'small' and 'large' based on parameter count, we aim to systematically assess how model scale interacts with robustness and accuracy under various perturbation regimes. Recent models, such as `Llama-3` and `Mistral`, demonstrate architectural innovations, including improved token representations and mixture-of-experts routing, which provide a richer set of inductive biases for our evaluation. Such a comprehensive suite enables a detailed comparison of robustness across architecture, scale, and fine-tuning strate-

| Model | Size | Date Released |
|---|---|---|
| **Small Model**(Less than 10B parameter) | | |
| Llama-2-7b-hf | 6.74B | July 2023 |
| Llama-2-7b-chat-hf | 6.74B | July 2023 |
| Mistral-7B-v0.1 | 7.24B | Sept 2023 |
| Mistral-7B-Instruct-v0.1 | 7.24B | Sept 2023 |
| Meta-Llama-3-8B | 8.03B | April 2024 |
| Meta-Llama-3-8B-Instruct | 8.03B | April 2024 |
| Qwen2.5-7B | 7.62B | Sept 2024 |
| Qwen2.5-7B-Instruct | 7.62B | Sept 2024 |
| Qwen3-8B-Base | 8.19B | April 2025 |
| Qwen3-8B | 8.19B | April 2025 |
| **Large Model**(Larger than 2s0B parameter) | | |
| Llama-2-70b-hf | 69B | July 2023 |
| Llama-2-70b-chat-hf | 69B | July 2023 |
| Mixtral-8x7B-v0.1 | 46.7B | Dec 2023 |
| Mixtral-8x7B-Instruct-v0.1 | 46.7B | Dec 2023 |
| Meta-Llama-3-70B | 70.6B | April 2024 |
| Meta-Llama-3-70B-Instruct | 70.6B | April 2024 |
| Qwen2.5-72B | 72.7B | Sept 2024 |
| Qwen2.5-72B-Instruct | 72.7B | Sept 2024 |
| Qwen3-30B-A3B-Base | 30.5B | April 2025 |
| Qwen3-30B-A3B | 30.5B | April 2025 |

Table 3: All the models with their parameter size and their date released. Large Models are defined as models with parameters larger than 40 billion parameters, and Small Models are models with parameters smaller than 10 billion parameters.

gies, thereby supporting more generalizable insights into LLM behavior on TQA tasks.

## F    Performance over Table Complexity

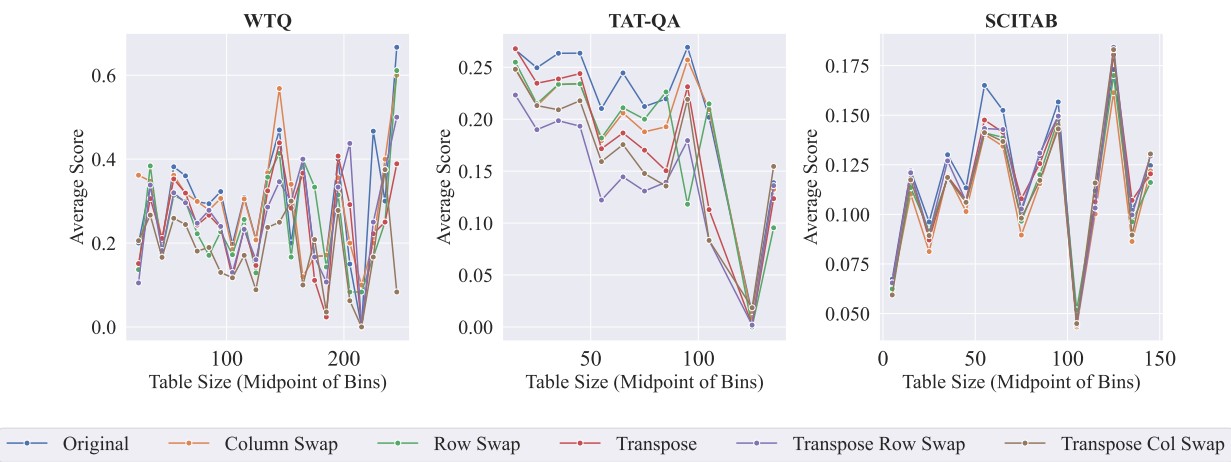

Figure 13: The average EM score for the different structural perturbations over the different sizes of tables on **WTQ**, **TAT-QA**, and **SCITAB** datasets

Figure 13 presents average model performance across varying table sizes for three tabular question-answering datasets—**WTQ**, **TAT-QA**, and **SCITAB**—under several table structure perturbations. Each plot compares the original condition with different table augmentation operations, such as column swapping, row swapping, and transpose-based modifications.

For **WTQ**, performance is low overall and highly variable across increasing table sizes, with no single condition consistently outperforming others. In contrast, **TAT-QA** shows a generally higher and more stable baseline, though still affected by growing table size and perturbations; performance tends to decline as tables grow larger, suggesting sensitivity to complexity. **SCITAB** results are somewhat mixed, with fluctuations at different size intervals, but the performance remains closer among the different conditions.

Across all three datasets, these results highlight that large language models, though capable, display uneven robustness to structural manipulations of tables. Although instruction tuning and larger model scales improve performance, structural changes continue to pose challenges, underscoring the need for more robust, structure-aware approaches to ensure reliable table comprehension.

## G    Additional Robustness Result

Across all three datasets—**WTQ**, **TAT-QA**, and **SCITAB**—the EM results in Table 4 elaborate on the consistent trends supporting the main findings. In addition to standard deviation for statistical significance, it also includes the TAPAS models(ranging from Tiny to Large) that are fine-tuned on the WTQ dataset and TAPEX models(Large and Base) fine-tuned on **WTQ** dataset. The TAPAS models exhibit a distinctive performance profile that contrasts sharply with the general-purpose LLMs evaluated in the main text. On WTQ, TAPAS achieves notably high EM scores on the original setting (0.562–0.720) and retains relatively strong performance under most structural perturbations, with only modest declines for operations such as column swap and row swap. However, its robustness sharply deteriorates under value-based perturbations such as DVP, RVP, and NVP, with EM drops similar to those seen for many instruction-tuned LLMs. On **TAT-QA**, TAPAS's performance is substantially lower than on **WTQ**, with EM scores often under 0.05 for most structural perturbations—an order-of-magnitude drop that reflects limited domain transferability from its trained dataset. Overall, TAPAS models show a better structural robustness within their training domain (**WTQ**) but exhibit steep cross-domain degradation and heightened vulnerability to semantic content

Table 4: EM results across all datasets (**WTQ**, **TAT-QA**, and **SCITAB**) and model families. Note: The labels **B**, **C**, and **I** refer to the Base, Chat, and Instruct variants of the model.

| Model | Original | Column Swap | Row Swap | Transpose | Transpose Row Swap | Transpose Col Swap | NT | DVP | RVP | NVP |
|---|---|---|---|---|---|---|---|---|---|---|
| **WTQ** | | | | | | | | | | |
| Llama2 7B | $0.15 \pm 0.01$ | $0.148 \pm 0.01$ | $0.118 \pm 0.018$ | $0.11 \pm 0.012$ | $0.108 \pm 0.015$ | $0.078 \pm 0.022$ | $0.021 \pm 0.003$ | $0.033 \pm 0.004$ | $0.012 \pm 0.008$ | $0.014 \pm 0.007$ |
| Llama3 8B | $0.27 \pm 0.035$ | $0.225 \pm 0.008$ | $0.221 \pm 0.0$ | $0.216 \pm 0.0$ | $0.181 \pm 0.015$ | $0.167 \pm 0.014$ | $0.016 \pm 0.0$ | $0.149 \pm 0.03$ | $0.085 \pm 0.0$ | $0.021 \pm 0.0$ |
| Mistral 7B | $0.294 \pm 0.0$ | $0.291 \pm 0.006$ | $0.224 \pm 0.006$ | $0.234 \pm 0.015$ | $0.214 \pm 0.003$ | $0.194 \pm 0.007$ | $0.025 \pm 0.0$ | $0.118 \pm 0.004$ | $0.066 \pm 0.018$ | $0.031 \pm 0.004$ |
| Qwen2.5 7B | $0.191 \pm 0.013$ | $0.196 \pm 0.025$ | $0.155 \pm 0.012$ | $0.158 \pm 0.003$ | $0.16 \pm 0.006$ | $0.131 \pm 0.02$ | $0.023 \pm 0.011$ | $0.064 \pm 0.014$ | $0.009 \pm 0.004$ | $0.026 \pm 0.022$ |
| Qwen3 8B **B** | $0.297 \pm 0.02$ | $0.246 \pm 0.03$ | $0.187 \pm 0.02$ | $0.272 \pm 0.015$ | $0.239 \pm 0.012$ | $0.168 \pm 0.012$ | $0.026 \pm 0.016$ | $0.139 \pm 0.018$ | $0.057 \pm 0.025$ | $0.031 \pm 0.004$ |
| Llama2 70B | $0.273 \pm 0.007$ | $0.299 \pm 0.013$ | $0.26 \pm 0.015$ | $0.235 \pm 0.032$ | $0.229 \pm 0.01$ | $0.178 \pm 0.007$ | $0.044 \pm 0.013$ | $0.135 \pm 0.021$ | $0.061 \pm 0.004$ | $0.038 \pm 0.008$ |
| Llama3 70B | $0.368 \pm 0.031$ | $0.36 \pm 0.04$ | $0.311 \pm 0.032$ | $0.348 \pm 0.0$ | $0.319 \pm 0.021$ | $0.25 \pm 0.04$ | $0.082 \pm 0.01$ | $0.199 \pm 0.021$ | $0.125 \pm 0.027$ | $0.085 \pm 0.012$ |
| Mistral 8x7B | $0.281 \pm 0.02$ | $0.291 \pm 0.012$ | $0.242 \pm 0.01$ | $0.265 \pm 0.013$ | $0.255 \pm 0.008$ | $0.219 \pm 0.011$ | $0.034 \pm 0.0$ | $0.147 \pm 0.004$ | $0.073 \pm 0.004$ | $0.05 \pm 0.0$ |
| Qwen2.5 72B | $0.37 \pm 0.021$ | $0.369 \pm 0.03$ | $0.306 \pm 0.019$ | $0.366 \pm 0.041$ | $0.375 \pm 0.016$ | $0.235 \pm 0.013$ | $0.02 \pm 0.0$ | $0.227 \pm 0.012$ | $0.137 \pm 0.015$ | $0.043 \pm 0.007$ |
| Qwen3 30B **B** | $0.265 \pm 0.01$ | $0.237 \pm 0.025$ | $0.225 \pm 0.0$ | $0.252 \pm 0.014$ | $0.209 \pm 0.014$ | $0.163 \pm 0.011$ | $0.015 \pm 0.008$ | $0.13 \pm 0.02$ | $0.076 \pm 0.011$ | $0.035 \pm 0.007$ |
| Llama2 7B **C** | $0.225 \pm 0.014$ | $0.233 \pm 0.003$ | $0.176 \pm 0.014$ | $0.181 \pm 0.0$ | $0.142 \pm 0.007$ | $0.123 \pm 0.007$ | $0.029 \pm 0.006$ | $0.121 \pm 0.0$ | $0.064 \pm 0.01$ | $0.011 \pm 0.005$ |
| Llama3 8B **I** | $0.373 \pm 0.013$ | $0.348 \pm 0.005$ | $0.33 \pm 0.019$ | $0.315 \pm 0.02$ | $0.314 \pm 0.013$ | $0.242 \pm 0.019$ | $0.01 \pm 0.01$ | $0.253 \pm 0.022$ | $0.168 \pm 0.011$ | $0.045 \pm 0.008$ |
| Mistral 7B **I** | $0.325 \pm 0.022$ | $0.324 \pm 0.0$ | $0.248 \pm 0.007$ | $0.284 \pm 0.018$ | $0.275 \pm 0.005$ | $0.189 \pm 0.01$ | $0.015 \pm 0.0$ | $0.248 \pm 0.007$ | $0.229 \pm 0.008$ | $0.019 \pm 0.004$ |
| Qwen2.5 7B **I** | $0.335 \pm 0.012$ | $0.312 \pm 0.01$ | $0.292 \pm 0.02$ | $0.273 \pm 0.025$ | $0.242 \pm 0.012$ | $0.19 \pm 0.012$ | $0.002 \pm 0.003$ | $0.236 \pm 0.03$ | $0.158 \pm 0.015$ | $0.057 \pm 0.007$ |
| Qwen3 8B | $0.295 \pm 0.017$ | $0.246 \pm 0.03$ | $0.187 \pm 0.02$ | $0.272 \pm 0.015$ | $0.239 \pm 0.012$ | $0.168 \pm 0.012$ | $0.01 \pm 0.013$ | $0.139 \pm 0.018$ | $0.057 \pm 0.025$ | $0.031 \pm 0.004$ |
| Llama2 70B **C** | $0.306 \pm 0.02$ | $0.302 \pm 0.017$ | $0.248 \pm 0.01$ | $0.25 \pm 0.008$ | $0.253 \pm 0.011$ | $0.201 \pm 0.018$ | $0.03 \pm 0.0$ | $0.187 \pm 0.022$ | $0.109 \pm 0.027$ | $0.045 \pm 0.004$ |
| Llama3 70B **I** | $0.544 \pm 0.01$ | $0.533 \pm 0.006$ | $0.447 \pm 0.015$ | $0.528 \pm 0.016$ | $0.497 \pm 0.012$ | $0.387 \pm 0.013$ | $0.083 \pm 0.005$ | $0.407 \pm 0.015$ | $0.314 \pm 0.018$ | $0.095 \pm 0.011$ |
| Mistral 8x7B **I** | $0.338 \pm 0.015$ | $0.317 \pm 0.006$ | $0.273 \pm 0.015$ | $0.327 \pm 0.022$ | $0.279 \pm 0.005$ | $0.225 \pm 0.013$ | $0.049 \pm 0.0$ | $0.149 \pm 0.007$ | $0.097 \pm 0.004$ | $0.061 \pm 0.004$ |
| Qwen2.5 72B **I** | $0.499 \pm 0.016$ | $0.501 \pm 0.023$ | $0.422 \pm 0.013$ | $0.472 \pm 0.02$ | $0.487 \pm 0.008$ | $0.365 \pm 0.016$ | $0.062 \pm 0.003$ | $0.369 \pm 0.007$ | $0.191 \pm 0.007$ | $0.061 \pm 0.008$ |
| Qwen3 30B | $0.261 \pm 0.012$ | $0.225 \pm 0.022$ | $0.204 \pm 0.03$ | $0.24 \pm 0.027$ | $0.209 \pm 0.01$ | $0.173 \pm 0.003$ | $0.018 \pm 0.019$ | $0.137 \pm 0.022$ | $0.083 \pm 0.015$ | $0.038 \pm 0.004$ |
| TAPAS Tiny (WTQ) | 0.562 | 0.507 | 0.230 | 0.078 | 0.064 | 0.078 | – | 0.355 | 0.418 | 0.064 |
| TAPAS Small (WTQ) | 0.700 | 0.691 | 0.389 | 0.078 | 0.083 | 0.049 | – | 0.645 | 0.603 | 0.064 |
| TAPAS Mini (WTQ) | 0.700 | 0.666 | 0.398 | 0.113 | 0.078 | 0.059 | – | 0.596 | 0.610 | 0.071 |
| TAPAS Medium (WTQ) | 0.720 | 0.686 | 0.478 | 0.074 | 0.059 | 0.069 | – | 0.695 | 0.624 | 0.057 |
| TAPAS Base (WTQ) | 0.705 | 0.696 | 0.465 | 0.083 | 0.044 | 0.088 | – | 0.695 | 0.603 | 0.050 |
| TAPAS Large (WTQ) | 0.713 | 0.700 | 0.485 | 0.116 | 0.096 | 0.103 | – | 0.702 | 0.596 | 0.057 |
| TAPEX Base (WTQ) | 0.000 | 0.000 | 0.000 | 0.000 | 0.000 | 0.000 | – | 0.000 | 0.000 | 0.000 |
| TAPEX Large (WTQ) | 0.098 | 0.078 | 0.079 | 0.020 | 0.034 | 0.025 | – | 0.078 | 0.057 | 0.000 |
| **TATQ-A** | | | | | | | | | | |
| Llama2 7B | $0.184 \pm 0.007$ | $0.16 \pm 0.008$ | $0.154 \pm 0.005$ | $0.147 \pm 0.001$ | $0.119 \pm 0.007$ | $0.133 \pm 0.001$ | $0.004 \pm 0.001$ | – | – | – |
| Llama3 8B | $0.298 \pm 0.002$ | $0.248 \pm 0.005$ | $0.26 \pm 0.004$ | $0.279 \pm 0.006$ | $0.218 \pm 0.014$ | $0.252 \pm 0.006$ | $0.003 \pm 0.0$ | – | – | – |
| Mistral 7B | $0.27 \pm 0.003$ | $0.237 \pm 0.001$ | $0.25 \pm 0.006$ | $0.239 \pm 0.004$ | $0.186 \pm 0.002$ | $0.216 \pm 0.003$ | $0.008 \pm 0.0$ | – | – | – |
| Qwen2.5 7B | $0.226 \pm 0.004$ | $0.195 \pm 0.006$ | $0.197 \pm 0.007$ | $0.216 \pm 0.005$ | $0.181 \pm 0.013$ | $0.208 \pm 0.004$ | $0.003 \pm 0.003$ | – | – | – |
| Qwen3 8B **B** | $0.254 \pm 0.005$ | $0.229 \pm 0.017$ | $0.218 \pm 0.002$ | $0.236 \pm 0.007$ | $0.208 \pm 0.012$ | $0.216 \pm 0.009$ | $0.0 \pm 0.0$ | – | – | – |
| Llama2 70B | $0.314 \pm 0.016$ | $0.283 \pm 0.022$ | $0.275 \pm 0.007$ | $0.263 \pm 0.01$ | $0.222 \pm 0.019$ | $0.267 \pm 0.008$ | $0.005 \pm 0.003$ | – | – | – |
| Llama3 70B | $0.384 \pm 0.011$ | $0.36 \pm 0.012$ | $0.362 \pm 0.015$ | $0.357 \pm 0.016$ | $0.322 \pm 0.004$ | $0.339 \pm 0.01$ | $0.014 \pm 0.001$ | – | – | – |
| Mistral 8x7B | $0.317 \pm 0.015$ | $0.279 \pm 0.002$ | $0.273 \pm 0.003$ | $0.284 \pm 0.01$ | $0.215 \pm 0.011$ | $0.275 \pm 0.004$ | $0.01 \pm 0.0$ | – | – | – |
| Qwen2.5 72B | $0.33 \pm 0.002$ | $0.3 \pm 0.008$ | $0.306 \pm 0.005$ | $0.305 \pm 0.003$ | $0.268 \pm 0.024$ | $0.307 \pm 0.012$ | $0.003 \pm 0.003$ | – | – | – |
| Qwen3 30B **B** | $0.336 \pm 0.011$ | $0.302 \pm 0.019$ | $0.304 \pm 0.004$ | $0.32 \pm 0.026$ | $0.239 \pm 0.008$ | $0.28 \pm 0.008$ | $0.006 \pm 0.003$ | – | – | – |
| Llama2 7B **C** | $0.185 \pm 0.004$ | $0.159 \pm 0.003$ | $0.168 \pm 0.007$ | $0.155 \pm 0.005$ | $0.124 \pm 0.003$ | $0.135 \pm 0.003$ | $0.008 \pm 0.004$ | – | – | – |
| Llama3 8B **I** | $0.278 \pm 0.004$ | $0.238 \pm 0.002$ | $0.253 \pm 0.004$ | $0.268 \pm 0.006$ | $0.215 \pm 0.007$ | $0.247 \pm 0.0$ | $0.001 \pm 0.001$ | – | – | – |
| Mistral 7B **I** | $0.229 \pm 0.006$ | $0.188 \pm 0.005$ | $0.197 \pm 0.004$ | $0.208 \pm 0.002$ | $0.17 \pm 0.003$ | $0.191 \pm 0.002$ | $0.001 \pm 0.0$ | – | – | – |
| Qwen2.5 7B **I** | $0.24 \pm 0.007$ | $0.222 \pm 0.017$ | $0.219 \pm 0.005$ | $0.227 \pm 0.003$ | $0.197 \pm 0.011$ | $0.218 \pm 0.009$ | $0.0 \pm 0.0$ | – | – | – |
| Qwen3 8B | $0.253 \pm 0.006$ | $0.241 \pm 0.009$ | $0.257 \pm 0.006$ | $0.249 \pm 0.008$ | $0.216 \pm 0.001$ | $0.226 \pm 0.018$ | $0.007 \pm 0.004$ | – | – | – |
| Llama2 70B **C** | $0.25 \pm 0.01$ | $0.211 \pm 0.008$ | $0.226 \pm 0.004$ | $0.211 \pm 0.006$ | $0.172 \pm 0.003$ | $0.197 \pm 0.012$ | $0.002 \pm 0.002$ | – | – | – |
| Llama3 70B **I** | $0.441 \pm 0.004$ | $0.417 \pm 0.006$ | $0.405 \pm 0.005$ | $0.431 \pm 0.001$ | $0.37 \pm 0.006$ | $0.391 \pm 0.005$ | $0.014 \pm 0.001$ | – | – | – |
| Mistral 8x7B **I** | $0.317 \pm 0.008$ | $0.283 \pm 0.017$ | $0.273 \pm 0.004$ | $0.27 \pm 0.006$ | $0.22 \pm 0.01$ | $0.271 \pm 0.013$ | $0.013 \pm 0.0$ | – | – | – |
| Qwen2.5 72B **I** | $0.353 \pm 0.009$ | $0.353 \pm 0.005$ | $0.337 \pm 0.006$ | $0.34 \pm 0.012$ | $0.303 \pm 0.006$ | $0.33 \pm 0.007$ | $0.008 \pm 0.0$ | – | – | – |
| Qwen3 30B | $0.346 \pm 0.022$ | $0.305 \pm 0.013$ | $0.304 \pm 0.027$ | $0.298 \pm 0.018$ | $0.258 \pm 0.008$ | $0.26 \pm 0.016$ | $0.01 \pm 0.0$ | – | – | – |
| TAPAS Tiny (WTQ) | 0.017 | 0.012 | 0.010 | 0.040 | 0.025 | 0.035 | – | – | – | – |
| TAPAS Small (WTQ) | 0.032 | 0.030 | 0.022 | 0.093 | 0.052 | 0.092 | – | – | – | – |
| TAPAS Mini (WTQ) | 0.045 | 0.046 | 0.015 | 0.105 | 0.057 | 0.097 | – | – | – | – |
| TAPAS Medium (WTQ) | 0.046 | 0.046 | 0.019 | 0.125 | 0.075 | 0.132 | – | – | – | – |
| TAPAS Base (WTQ) | 0.039 | 0.040 | 0.021 | 0.071 | 0.054 | 0.065 | – | – | – | – |
| TAPAS Large (WTQ) | 0.037 | 0.045 | 0.029 | 0.085 | 0.058 | 0.087 | – | – | – | – |
| TAPEX Base (WTQ) | 0.010 | 0.020 | 0.008 | 0.012 | 0.012 | 0.016 | – | – | – | – |
| TAPEX Large (WTQ) | 0.012 | 0.008 | 0.005 | 0.002 | 0.008 | 0.009 | – | – | – | – |
| **SCITAB** | | | | | | | | | | |
| Llama2 7B | $0.001 \pm 0.001$ | $0.003 \pm 0.004$ | $0.002 \pm 0.0$ | $0.002 \pm 0.001$ | $0.002 \pm 0.003$ | $0.001 \pm 0.001$ | $0.0 \pm 0.0$ | – | – | – |
| Llama3 8B | $0.338 \pm 0.025$ | $0.349 \pm 0.03$ | $0.32 \pm 0.016$ | $0.342 \pm 0.011$ | $0.352 \pm 0.03$ | $0.319 \pm 0.017$ | $0.0 \pm 0.0$ | – | – | – |
| Mistral 7B | $0.0 \pm 0.0$ | $0.0 \pm 0.0$ | $0.0 \pm 0.0$ | $0.0 \pm 0.0$ | $0.0 \pm 0.0$ | $0.0 \pm 0.0$ | $0.0 \pm 0.0$ | – | – | – |
| Qwen2.5 7B | $0.002 \pm 0.004$ | $0.0 \pm 0.0$ | $0.002 \pm 0.001$ | $0.002 \pm 0.003$ | $0.003 \pm 0.004$ | $0.002 \pm 0.001$ | $0.0 \pm 0.0$ | – | – | – |
| Qwen3 8B **B** | $0.327 \pm 0.036$ | $0.293 \pm 0.013$ | $0.301 \pm 0.019$ | $0.312 \pm 0.011$ | $0.28 \pm 0.007$ | $0.278 \pm 0.017$ | $0.0 \pm 0.0$ | – | – | – |
| Llama2 70B | $0.034 \pm 0.007$ | $0.027 \pm 0.028$ | $0.022 \pm 0.02$ | $0.04 \pm 0.009$ | $0.036 \pm 0.013$ | $0.031 \pm 0.008$ | $0.0 \pm 0.0$ | – | – | – |
| Llama3 70B | $0.38 \pm 0.03$ | $0.366 \pm 0.007$ | $0.35 \pm 0.009$ | $0.367 \pm 0.007$ | $0.364 \pm 0.007$ | $0.356 \pm 0.043$ | $0.0 \pm 0.0$ | – | – | – |
| Mistral 8x7B | $0.0 \pm 0.0$ | $0.0 \pm 0.0$ | $0.0 \pm 0.0$ | $0.0 \pm 0.0$ | $0.0 \pm 0.0$ | $0.0 \pm 0.0$ | $0.0 \pm 0.0$ | – | – | – |
| Qwen2.5 72B | $0.349 \pm 0.019$ | $0.3 \pm 0.035$ | $0.293 \pm 0.045$ | $0.341 \pm 0.036$ | $0.311 \pm 0.005$ | $0.359 \pm 0.058$ | $0.0 \pm 0.0$ | – | – | – |
| Qwen3 30B **B** | $0.093 \pm 0.012$ | $0.065 \pm 0.011$ | $0.074 \pm 0.01$ | $0.082 \pm 0.009$ | $0.078 \pm 0.007$ | $0.059 \pm 0.01$ | $0.0 \pm 0.0$ | – | – | – |
| Llama2 7B **C** | $0.384 \pm 0.006$ | $0.387 \pm 0.008$ | $0.382 \pm 0.0$ | $0.378 \pm 0.005$ | $0.383 \pm 0.008$ | $0.387 \pm 0.004$ | $0.0 \pm 0.0$ | – | – | – |
| Llama3 8B **I** | $0.32 \pm 0.003$ | $0.313 \pm 0.017$ | $0.322 \pm 0.008$ | $0.315 \pm 0.014$ | $0.324 \pm 0.004$ | $0.311 \pm 0.009$ | $0.0 \pm 0.0$ | – | – | – |
| Mistral 7B **I** | $0.284 \pm 0.012$ | $0.28 \pm 0.016$ | $0.28 \pm 0.013$ | $0.282 \pm 0.018$ | $0.279 \pm 0.009$ | $0.283 \pm 0.011$ | $0.0 \pm 0.0$ | – | – | – |
| Qwen2.5 7B **I** | $0.352 \pm 0.003$ | $0.322 \pm 0.007$ | $0.353 \pm 0.013$ | $0.364 \pm 0.01$ | $0.347 \pm 0.004$ | $0.339 \pm 0.004$ | $0.0 \pm 0.0$ | – | – | – |
| Qwen3 8B | $0.378 \pm 0.016$ | $0.341 \pm 0.013$ | $0.325 \pm 0.011$ | $0.378 \pm 0.034$ | $0.356 \pm 0.005$ | $0.322 \pm 0.009$ | $0.0 \pm 0.0$ | – | – | – |
| Llama2 70B **C** | $0.084 \pm 0.004$ | $0.072 \pm 0.01$ | $0.076 \pm 0.005$ | $0.091 \pm 0.013$ | $0.077 \pm 0.008$ | $0.078 \pm 0.014$ | $0.0 \pm 0.0$ | – | – | – |
| Llama3 70B **C** | $0.475 \pm 0.007$ | $0.462 \pm 0.014$ | $0.457 \pm 0.006$ | $0.461 \pm 0.014$ | $0.454 \pm 0.017$ | $0.451 \pm 0.014$ | $0.0 \pm 0.0$ | – | – | – |
| Mistral 8x7B **I** | $0.083 \pm 0.001$ | $0.078 \pm 0.008$ | $0.08 \pm 0.004$ | $0.084 \pm 0.007$ | $0.087 \pm 0.008$ | $0.082 \pm 0.003$ | $0.0 \pm 0.0$ | – | – | – |
| Qwen2.5 72B **I** | $0.63 \pm 0.012$ | $0.578 \pm 0.011$ | $0.579 \pm 0.01$ | $0.626 \pm 0.023$ | $0.575 \pm 0.011$ | $0.6 \pm 0.014$ | $0.0 \pm 0.0$ | – | – | – |
| Qwen3 30B | $0.021 \pm 0.008$ | $0.022 \pm 0.012$ | $0.012 \pm 0.001$ | $0.008 \pm 0.0$ | $0.015 \pm 0.002$ | $0.007 \pm 0.004$ | $0.0 \pm 0.0$ | – | – | – |

changes, reinforcing the main-text conclusion that while specialized tabular models can excel in-domain, instruction-tuned LLMs demonstrate superior adaptability and broader robustness across perturbations and domains. Across all datasets, TAPEX performs substantially worse than both TAPAS and modern LLM-based models, with EM scores often near zero regardless of perturbation. TAPEX models — even the WTQ-specific large variant, peak at only 0.098 EM and collapse under perturbations, indicating a lack of generalization, likely due to training, evaluation mismatches, or overfitting. On TAT-QA, LLM-based models again lead (e.g., Llama3-70B I at 0.441), whereas TAPAS drops to the 0.017–0.046 range and TAPEX remains near zero.

## G.1 Score with F1 Measure

Table 5: F1 measure results across all datasets (**WTQ**, **TAT-QA**, and **SCITAB**) and model families. Note: The labels **B**, **C**, and **I** refer to the Base, Chat, and Instruct variants of the model.

| Model | Original | Column Swap | Row Swap | Transpose | Transpose Row Swap | Transpose Col Swap | NT | DVP | RVP | NVP |
|---|---|---|---|---|---|---|---|---|---|---|
| **WTQ** | | | | | | | | | | |
| Llama2 7B | 0.173 ± 0.008 | 0.169 ± 0.01 | 0.134 ± 0.014 | 0.127 ± 0.009 | 0.129 ± 0.014 | 0.09 ± 0.021 | 0.032 ± 0.004 | 0.053 ± 0.01 | 0.022 ± 0.012 | 0.028 ± 0.007 |
| Llama3 8B | 0.316 ± 0.021 | 0.254 ± 0.012 | 0.252 ± 0.0 | 0.259 ± 0.0 | 0.212 ± 0.02 | 0.193 ± 0.014 | 0.027 ± 0.002 | 0.187 ± 0.035 | 0.112 ± 0.0 | 0.027 ± 0.0 |
| Mistral 7B | 0.322 ± 0.008 | 0.321 ± 0.008 | 0.252 ± 0.004 | 0.262 ± 0.01 | 0.243 ± 0.003 | 0.214 ± 0.006 | 0.037 ± 0.005 | 0.139 ± 0.004 | 0.083 ± 0.019 | 0.036 ± 0.002 |
| Qwen2.5 7B | 0.271 ± 0.015 | 0.254 ± 0.017 | 0.206 ± 0.014 | 0.206 ± 0.005 | 0.212 ± 0.006 | 0.161 ± 0.019 | 0.042 ± 0.011 | 0.111 ± 0.011 | 0.044 ± 0.004 | 0.04 ± 0.023 |
| Qwen3 8B **B** | 0.35 ± 0.01 | 0.302 ± 0.03 | 0.252 ± 0.021 | 0.318 ± 0.021 | 0.286 ± 0.015 | 0.191 ± 0.013 | 0.043 ± 0.022 | 0.173 ± 0.016 | 0.091 ± 0.03 | 0.039 ± 0.005 |
| Llama2 70B | 0.308 ± 0.006 | 0.331 ± 0.013 | 0.292 ± 0.011 | 0.268 ± 0.032 | 0.246 ± 0.011 | 0.2 ± 0.007 | 0.038 ± 0.009 | 0.15 ± 0.023 | 0.079 ± 0.007 | 0.054 ± 0.014 |
| Llama3 70B | 0.414 ± 0.036 | 0.4 ± 0.033 | 0.354 ± 0.03 | 0.38 ± 0.008 | 0.352 ± 0.021 | 0.277 ± 0.038 | 0.105 ± 0.01 | 0.217 ± 0.026 | 0.139 ± 0.022 | 0.109 ± 0.01 |
| Mistral 8x7B | 0.305 ± 0.023 | 0.313 ± 0.012 | 0.258 ± 0.004 | 0.291 ± 0.012 | 0.275 ± 0.004 | 0.239 ± 0.009 | 0.04 ± 0.001 | 0.162 ± 0.006 | 0.082 ± 0.005 | 0.061 ± 0.005 |
| Qwen2.5 72B | 0.432 ± 0.019 | 0.426 ± 0.024 | 0.364 ± 0.018 | 0.415 ± 0.04 | 0.431 ± 0.011 | 0.27 ± 0.008 | 0.037 ± 0.002 | 0.25 ± 0.015 | 0.159 ± 0.02 | 0.069 ± 0.007 |
| Qwen3 30B **B** | 0.313 ± 0.016 | 0.272 ± 0.013 | 0.275 ± 0.017 | 0.299 ± 0.014 | 0.247 ± 0.018 | 0.202 ± 0.01 | 0.045 ± 0.01 | 0.168 ± 0.025 | 0.108 ± 0.005 | 0.059 ± 0.009 |
| Llama2 7B **C** | 0.269 ± 0.008 | 0.271 ± 0.008 | 0.204 ± 0.022 | 0.209 ± 0.004 | 0.171 ± 0.005 | 0.145 ± 0.01 | 0.057 ± 0.003 | 0.146 ± 0.01 | 0.078 ± 0.02 | 0.04 ± 0.006 |
| Llama3 8B **I** | 0.46 ± 0.014 | 0.434 ± 0.008 | 0.41 ± 0.027 | 0.386 ± 0.016 | 0.368 ± 0.007 | 0.292 ± 0.016 | 0.022 ± 0.016 | 0.336 ± 0.023 | 0.235 ± 0.004 | 0.072 ± 0.005 |
| Mistral 7B **I** | 0.417 ± 0.016 | 0.409 ± 0.004 | 0.343 ± 0.01 | 0.371 ± 0.016 | 0.351 ± 0.006 | 0.257 ± 0.016 | 0.025 ± 0.0 | 0.327 ± 0.004 | 0.289 ± 0.012 | 0.065 ± 0.003 |
| Qwen2.5 7B **I** | 0.459 ± 0.007 | 0.442 ± 0.009 | 0.397 ± 0.036 | 0.415 ± 0.023 | 0.366 ± 0.02 | 0.306 ± 0.011 | 0.026 ± 0.003 | 0.335 ± 0.026 | 0.233 ± 0.006 | 0.09 ± 0.011 |
| Qwen3 8B | 0.347 ± 0.011 | 0.302 ± 0.03 | 0.252 ± 0.021 | 0.318 ± 0.021 | 0.286 ± 0.015 | 0.191 ± 0.013 | 0.032 ± 0.014 | 0.173 ± 0.016 | 0.091 ± 0.03 | 0.039 ± 0.005 |
| Llama2 70B **C** | 0.339 ± 0.011 | 0.341 ± 0.01 | 0.282 ± 0.012 | 0.272 ± 0.005 | 0.281 ± 0.008 | 0.215 ± 0.033 | 0.024 ± 0.01 | 0.214 ± 0.02 | 0.134 ± 0.036 | 0.056 ± 0.007 |
| Llama3 70B **I** | 0.619 ± 0.016 | 0.605 ± 0.014 | 0.531 ± 0.007 | 0.611 ± 0.014 | 0.574 ± 0.024 | 0.466 ± 0.013 | 0.109 ± 0.007 | 0.491 ± 0.005 | 0.383 ± 0.018 | 0.13 ± 0.011 |
| Mistral 8x7B **I** | 0.406 ± 0.006 | 0.38 ± 0.008 | 0.319 ± 0.02 | 0.386 ± 0.02 | 0.334 ± 0.005 | 0.253 ± 0.012 | 0.057 ± 0.01 | 0.203 ± 0.014 | 0.113 ± 0.004 | 0.072 ± 0.008 |
| Qwen2.5 72B **I** | 0.589 ± 0.013 | 0.59 ± 0.01 | 0.511 ± 0.005 | 0.551 ± 0.012 | 0.563 ± 0.019 | 0.432 ± 0.004 | 0.091 ± 0.005 | 0.475 ± 0.004 | 0.265 ± 0.01 | 0.103 ± 0.011 |
| Qwen3 30B | 0.368 ± 0.025 | 0.345 ± 0.025 | 0.301 ± 0.029 | 0.34 ± 0.027 | 0.304 ± 0.008 | 0.245 ± 0.021 | 0.041 ± 0.017 | 0.24 ± 0.007 | 0.187 ± 0.01 | 0.064 ± 0.01 |
| TAPAS Tiny (WTQ) | 0.584 | 0.525 | 0.282 | 0.094 | 0.079 | 0.091 | – | 0.390 | 0.451 | 0.064 |
| TAPAS Small (WTQ) | 0.717 | 0.705 | 0.422 | 0.087 | 0.098 | 0.061 | – | 0.665 | 0.621 | 0.069 |
| TAPAS Mini (WTQ) | 0.717 | 0.683 | 0.423 | 0.120 | 0.087 | 0.074 | – | 0.635 | 0.633 | 0.082 |
| TAPAS Medium (WTQ) | 0.737 | 0.704 | 0.498 | 0.085 | 0.062 | 0.078 | – | 0.712 | 0.643 | 0.064 |
| TAPAS Base (WTQ) | 0.720 | 0.710 | 0.484 | 0.097 | 0.060 | 0.105 | – | 0.706 | 0.624 | 0.076 |
| TAPAS Large (WTQ) | 0.735 | 0.716 | 0.519 | 0.120 | 0.096 | 0.109 | – | 0.709 | 0.611 | 0.089 |
| TAPEX Base (WTQ) | 0.020 | 0.025 | 0.039 | 0.012 | 0.014 | 0.029 | – | 0.022 | 0.061 | 0.004 |
| TAPEX Large (WTQ) | 0.120 | 0.093 | 0.100 | 0.032 | 0.048 | 0.040 | – | 0.089 | 0.068 | 0.018 |
| **TATQ-A** | | | | | | | | | | |
| Llama2 7B | 0.216 ± 0.007 | 0.191 ± 0.01 | 0.185 ± 0.004 | 0.179 ± 0.001 | 0.146 ± 0.005 | 0.163 ± 0.004 | 0.014 ± 0.001 | – | – | – |
| Llama3 8B | 0.348 ± 0.003 | 0.294 ± 0.004 | 0.306 ± 0.002 | 0.326 ± 0.005 | 0.264 ± 0.014 | 0.297 ± 0.006 | 0.015 ± 0.002 | – | – | – |
| Mistral 7B | 0.329 ± 0.002 | 0.292 ± 0.001 | 0.306 ± 0.007 | 0.295 ± 0.003 | 0.235 ± 0.004 | 0.272 ± 0.004 | 0.021 ± 0.0 | – | – | – |
| Qwen2.5 7B | 0.282 ± 0.002 | 0.254 ± 0.009 | 0.251 ± 0.005 | 0.27 ± 0.008 | 0.233 ± 0.014 | 0.264 ± 0.003 | 0.02 ± 0.002 | – | – | – |
| Qwen3 8B **B** | 0.304 ± 0.005 | 0.28 ± 0.016 | 0.27 ± 0.005 | 0.285 ± 0.01 | 0.256 ± 0.016 | 0.266 ± 0.009 | 0.022 ± 0.001 | – | – | – |
| Llama2 70B | 0.378 ± 0.015 | 0.349 ± 0.022 | 0.34 ± 0.006 | 0.33 ± 0.01 | 0.283 ± 0.02 | 0.328 ± 0.011 | 0.013 ± 0.004 | – | – | – |
| Llama3 70B | 0.45 ± 0.011 | 0.423 ± 0.013 | 0.424 ± 0.01 | 0.421 ± 0.018 | 0.389 ± 0.002 | 0.403 ± 0.011 | 0.054 ± 0.0 | – | – | – |
| Mistral 8x7B | 0.364 ± 0.014 | 0.322 ± 0.004 | 0.319 ± 0.001 | 0.333 ± 0.01 | 0.256 ± 0.013 | 0.325 ± 0.003 | 0.031 ± 0.0 | – | – | – |
| Qwen2.5 72B | 0.388 ± 0.002 | 0.358 ± 0.002 | 0.361 ± 0.006 | 0.365 ± 0.008 | 0.327 ± 0.012 | 0.364 ± 0.018 | 0.02 ± 0.007 | – | – | – |
| Qwen3 30B **B** | 0.384 ± 0.014 | 0.351 ± 0.021 | 0.355 ± 0.004 | 0.371 ± 0.027 | 0.29 ± 0.008 | 0.328 ± 0.005 | 0.031 ± 0.002 | – | – | – |
| Llama2 7B **C** | 0.23 ± 0.006 | 0.198 ± 0.003 | 0.21 ± 0.008 | 0.199 ± 0.004 | 0.156 ± 0.005 | 0.175 ± 0.003 | 0.028 ± 0.012 | – | – | – |
| Llama3 8B **I** | 0.35 ± 0.006 | 0.305 ± 0.003 | 0.321 ± 0.0 | 0.34 ± 0.006 | 0.279 ± 0.006 | 0.311 ± 0.002 | 0.005 ± 0.002 | – | – | – |
| Mistral 7B **I** | 0.312 ± 0.002 | 0.261 ± 0.003 | 0.272 ± 0.002 | 0.286 ± 0.0 | 0.24 ± 0.002 | 0.267 ± 0.003 | 0.024 ± 0.0 | – | – | – |
| Qwen2.5 7B **I** | 0.306 ± 0.007 | 0.285 ± 0.015 | 0.281 ± 0.007 | 0.294 ± 0.003 | 0.26 ± 0.013 | 0.284 ± 0.009 | 0.018 ± 0.002 | – | – | – |
| Qwen3 8B | 0.316 ± 0.008 | 0.301 ± 0.008 | 0.321 ± 0.005 | 0.32 ± 0.009 | 0.279 ± 0.002 | 0.287 ± 0.017 | 0.033 ± 0.004 | – | – | – |
| Llama2 70B **C** | 0.307 ± 0.011 | 0.268 ± 0.009 | 0.282 ± 0.007 | 0.27 ± 0.007 | 0.221 ± 0.004 | 0.248 ± 0.008 | 0.004 ± 0.003 | – | – | – |
| Llama3 70B **I** | 0.523 ± 0.005 | 0.499 ± 0.005 | 0.486 ± 0.005 | 0.514 ± 0.001 | 0.452 ± 0.006 | 0.472 ± 0.005 | 0.054 ± 0.0 | – | – | – |
| Mistral 8x7B **I** | 0.372 ± 0.015 | 0.336 ± 0.016 | 0.327 ± 0.004 | 0.326 ± 0.007 | 0.269 ± 0.01 | 0.325 ± 0.011 | 0.036 ± 0.0 | – | – | – |
| Qwen2.5 72B **I** | 0.426 ± 0.008 | 0.426 ± 0.006 | 0.411 ± 0.009 | 0.418 ± 0.01 | 0.377 ± 0.004 | 0.4 ± 0.005 | 0.04 ± 0.0 | – | – | – |
| Qwen3 30B | 0.409 ± 0.019 | 0.373 ± 0.02 | 0.373 ± 0.035 | 0.371 ± 0.021 | 0.319 ± 0.01 | 0.329 ± 0.021 | 0.034 ± 0.0 | – | – | – |
| TAPAS Tiny (WTQ) | 0.031 | 0.026 | 0.026 | 0.047 | 0.035 | 0.046 | – | – | – | – |
| TAPAS Small (WTQ) | 0.050 | 0.043 | 0.031 | 0.100 | 0.060 | 0.099 | – | – | – | – |
| TAPAS Mini (WTQ) | 0.053 | 0.052 | 0.024 | 0.112 | 0.062 | 0.103 | – | – | – | – |
| TAPAS Medium (WTQ) | 0.057 | 0.061 | 0.032 | 0.145 | 0.084 | 0.144 | – | – | – | – |
| TAPAS Base (WTQ) | 0.052 | 0.049 | 0.033 | 0.083 | 0.063 | 0.081 | – | – | – | – |
| TAPAS Large (WTQ) | 0.045 | 0.052 | 0.039 | 0.106 | 0.074 | 0.115 | – | – | – | – |
| TAPEX Base (WTQ) | 0.013 | 0.021 | 0.010 | 0.013 | 0.013 | 0.020 | – | – | – | – |
| TAPEX Large (WTQ) | 0.020 | 0.017 | 0.012 | 0.009 | 0.013 | 0.016 | – | – | – | – |
| **SCITAB** | | | | | | | | | | |
| Llama2 7B | 0.001 ± 0.002 | 0.003 ± 0.004 | 0.002 ± 0.0 | 0.002 ± 0.001 | 0.002 ± 0.003 | 0.001 ± 0.002 | 0.0 ± 0.0 | – | – | – |
| Llama3 8B | 0.341 ± 0.024 | 0.351 ± 0.031 | 0.323 ± 0.016 | 0.345 ± 0.011 | 0.354 ± 0.03 | 0.32 ± 0.018 | 0.0 ± 0.0 | – | – | – |
| Mistral 7B | 0.0 ± 0.0 | 0.0 ± 0.0 | 0.0 ± 0.0 | 0.0 ± 0.0 | 0.0 ± 0.0 | 0.0 ± 0.0 | 0.0 ± 0.0 | | | |
| Qwen2.5 7B | 0.003 ± 0.004 | 0.001 ± 0.001 | 0.002 ± 0.001 | 0.003 ± 0.003 | 0.003 ± 0.004 | 0.002 ± 0.001 | 0.0 ± 0.0 | | | |
| Qwen3 8B **B** | 0.329 ± 0.036 | 0.297 ± 0.012 | 0.304 ± 0.022 | 0.318 ± 0.012 | 0.284 ± 0.008 | 0.283 ± 0.016 | 0.0 ± 0.0 | | | |
| Llama2 70B | 0.034 ± 0.007 | 0.027 ± 0.029 | 0.023 ± 0.02 | 0.042 ± 0.012 | 0.036 ± 0.014 | 0.032 ± 0.008 | 0.0 ± 0.0 | | | |
| Llama3 70B | 0.38 ± 0.029 | 0.366 ± 0.007 | 0.352 ± 0.009 | 0.369 ± 0.004 | 0.365 ± 0.007 | 0.356 ± 0.043 | 0.0 ± 0.0 | | | |
| Mistral 8x7B | 0.0 ± 0.0 | 0.0 ± 0.0 | 0.0 ± 0.0 | 0.0 ± 0.0 | 0.0 ± 0.0 | 0.0 ± 0.0 | 0.0 ± 0.0 | | | |
| Qwen2.5 72B | 0.353 ± 0.021 | 0.305 ± 0.031 | 0.295 ± 0.046 | 0.344 ± 0.038 | 0.315 ± 0.006 | 0.365 ± 0.064 | 0.0 ± 0.0 | | | |
| Qwen3 30B **B** | 0.105 ± 0.015 | 0.074 ± 0.005 | 0.085 ± 0.014 | 0.094 ± 0.011 | 0.086 ± 0.007 | 0.067 ± 0.005 | 0.0 ± 0.0 | | | |
| Llama2 7B **C** | 0.384 ± 0.007 | 0.387 ± 0.008 | 0.383 ± 0.0 | 0.378 ± 0.005 | 0.385 ± 0.008 | 0.387 ± 0.004 | 0.0 ± 0.0 | – | – | – |
| Llama3 8B **I** | 0.328 ± 0.004 | 0.322 ± 0.016 | 0.331 ± 0.008 | 0.322 ± 0.015 | 0.333 ± 0.003 | 0.32 ± 0.008 | 0.0 ± 0.0 | – | – | – |
| Mistral 7B **I** | 0.29 ± 0.012 | 0.286 ± 0.015 | 0.287 ± 0.01 | 0.285 ± 0.01 | 0.284 ± 0.009 | 0.288 ± 0.011 | 0.0 ± 0.0 | – | – | – |
| Qwen2.5 7B **I** | 0.371 ± 0.005 | 0.335 ± 0.007 | 0.369 ± 0.012 | 0.376 ± 0.01 | 0.359 ± 0.004 | 0.351 ± 0.004 | 0.0 ± 0.0 | – | – | – |
| Qwen3 8B | 0.383 ± 0.018 | 0.348 ± 0.011 | 0.332 ± 0.009 | 0.383 ± 0.034 | 0.364 ± 0.006 | 0.33 ± 0.007 | 0.0 ± 0.0 | – | – | – |
| Llama2 70B **C** | 0.084 ± 0.004 | 0.072 ± 0.01 | 0.076 ± 0.006 | 0.091 ± 0.012 | 0.077 ± 0.008 | 0.078 ± 0.014 | 0.0 ± 0.0 | – | – | – |
| Llama3 70B **I** | 0.478 ± 0.006 | 0.465 ± 0.016 | 0.46 ± 0.005 | 0.466 ± 0.014 | 0.458 ± 0.018 | 0.454 ± 0.014 | 0.0 ± 0.0 | – | – | – |
| Mistral 8x7B **I** | 0.083 ± 0.001 | 0.08 ± 0.009 | 0.08 ± 0.005 | 0.084 ± 0.007 | 0.087 ± 0.008 | 0.083 ± 0.003 | 0.0 ± 0.0 | – | – | – |
| Qwen2.5 72B **I** | 0.632 ± 0.014 | 0.585 ± 0.011 | 0.585 ± 0.012 | 0.628 ± 0.023 | 0.579 ± 0.007 | 0.603 ± 0.013 | 0.0 ± 0.0 | – | – | – |
| Qwen3 30B | 0.038 ± 0.009 | 0.03 ± 0.013 | 0.021 ± 0.003 | 0.022 ± 0.0 | 0.024 ± 0.004 | 0.015 ± 0.005 | 0.0 ± 0.0 | – | – | – |

The extended evaluation Table 5 expands the comparison to include the TAPAS family (Tiny to Large) model trained on the **WTQ** dataset. These results reaffirm and sharpen several robustness patterns identified in the main paper. First, newer architectures such as Llama3 and Qwen3 consistently achieve higher average F1 scores across all datasets and perturbations, with instruction-tuned variants showing the most consistent robustness gains. The Llama2 and Qwen2.5 families, while competitive on **WTQ**, exhibit larger relative performance drops when evaluated on out-of-domain datasets (**TAT-QA**, **SCITAB**), suggesting weaker cross-domain generalization.

Second, TAPAS demonstrates a markedly different profile. As a structure-aware, table-specific model, TAPAS achieves the highest in-domain (**WTQ**) performance under nearly all perturbations, particularly in column swap scenarios. However, its accuracy declines sharply on **TAT-QA**, confirming a robustness–generalization tradeoff: specialized models excel within their training domain but are more sensitive to schema and domain shifts. Across all model classes, column swap emerges as the least destructive structural perturbation, whereas transpose-based operations, especially transpose column swap, yield the steepest declines, consistent with the trends reported in the main study.

Since TAPAS is originally trained to solve synthetic SQL queries and is fine-tuned for table-based question answering and fact verification. Given that the **SCITAB** dataset can be formulated as a question-answering task, without fine-tuning on it, it is not possible to generate a scientific claims verification problem; hence, omitted for comparison.

TAPEX shows markedly better results under the relaxed F1 metric compared to its near-zero EM scores(Table 4), indicating that while it rarely produces exact matches, it often generates partially correct answers. On **WTQ**, the **WTQ** finetuned TAPEX Large variant, which achieved only 0.098 EM, improves substantially under F1, reflecting better token-level overlap despite structural or formatting mismatches. Similar patterns hold in **TAT-QA**, where EM performance was effectively zero, but F1 reveals non-trivial overlap with the gold answers, suggesting that TAPEX captures fragments of the correct information even when failing to match exactly. This gain under F1 highlights that TAPEX's weaknesses in these evaluations stem less from a complete lack of understanding and more from its inability to align outputs precisely with the expected format, reinforcing the importance of using both exact and relaxed metrics when assessing its capabilities.

# H   Attention Matrix Analysis

## H.1   Spearman Correlation within All Attention Heads

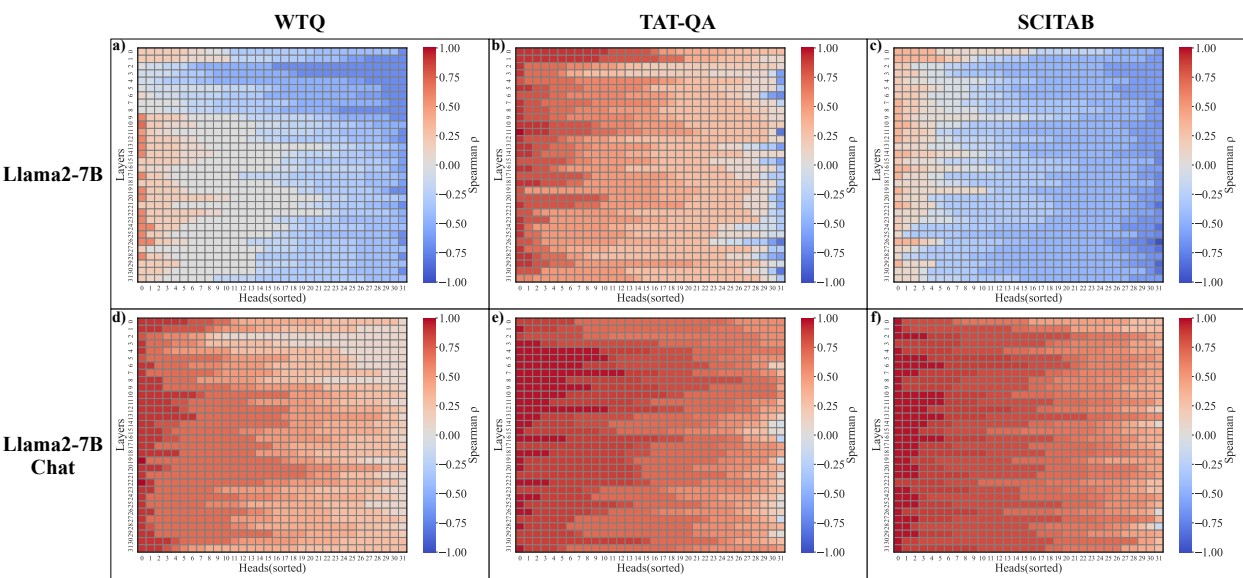

Figure 14: Heatmap showing Spearman correlation between changes in attention entropy and EM difference across all attention heads and layers in the `Llama2-7B` model(**a, b** and **c**) and `Llama2-7B-chat` model(**d, e** and **f**).

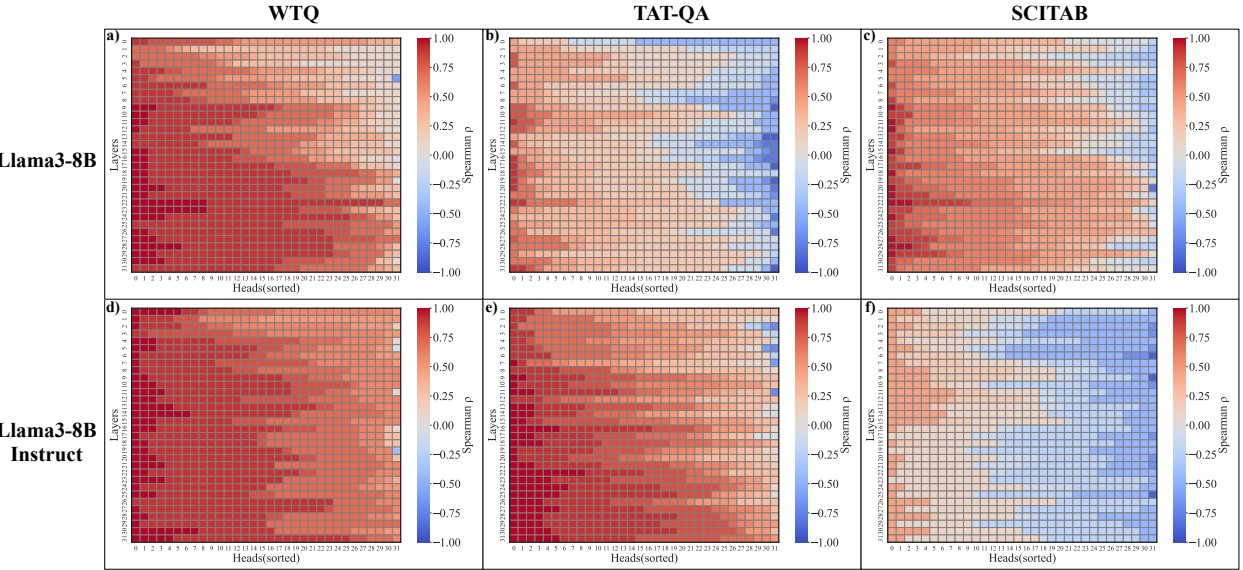

Figure 15: Heatmap showing Spearman correlation between changes in attention entropy and EM difference across all attention heads and layers in the `Llama3-8B` model(**a, b** and **c**) and `Llama3-8B-instruct` model(**d, e** and **f**).

Across both the model families `Llama3` and `Mistral`, the middle layers consistently exhibit the strongest positive Spearman correlations between perturbation-induced changes in attention entropy and EM degradation. This recurring 'hot spot' in the mid-layer shows a general architectural property; middle transformer blocks

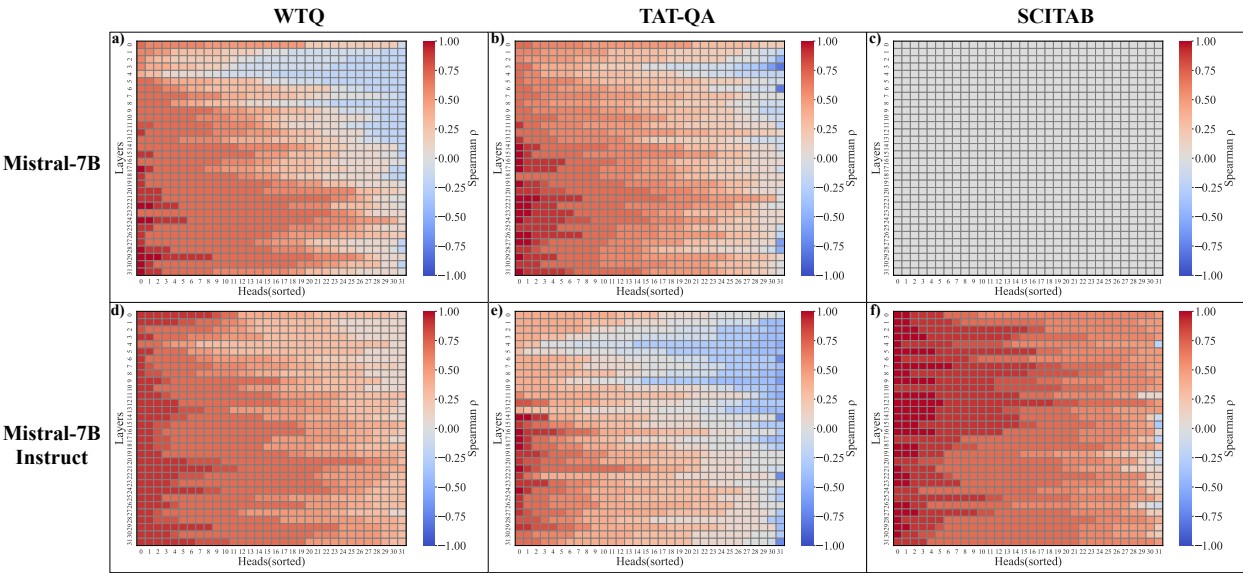

Figure 16: Heatmap showing Spearman correlation between changes in attention entropy and EM difference across all attention heads and layers in the `Mistral-7B` model(**a, b** and **c**) and `Mistral-7B-instruct` model(**d, e** and **f**).

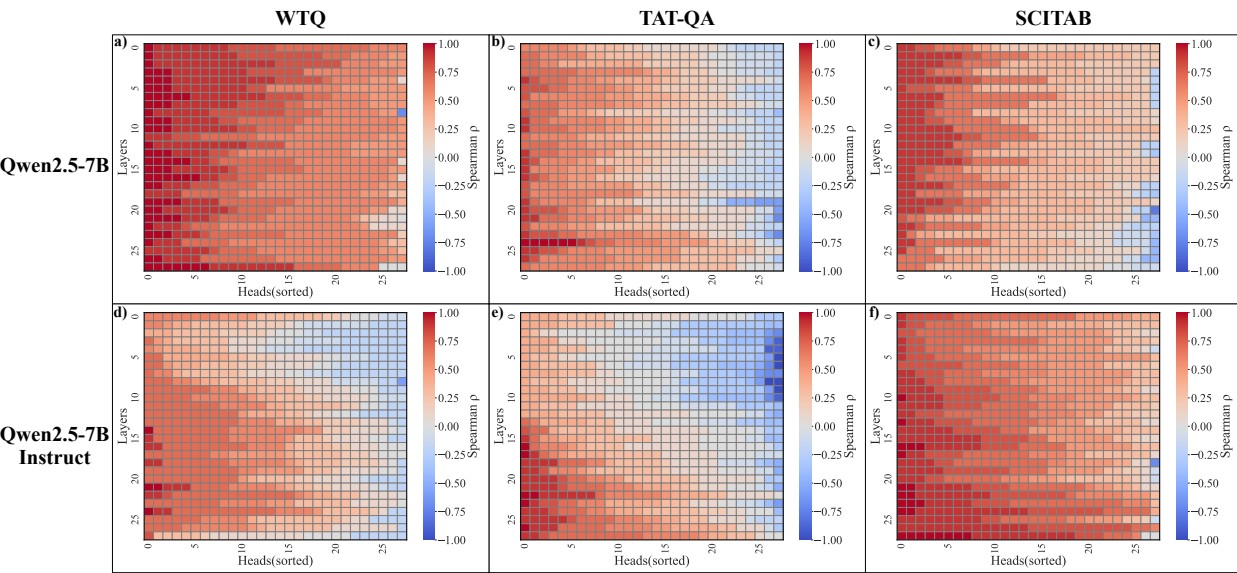

Figure 17: Heatmap showing Spearman correlation between changes in attention entropy and EM difference across all attention heads and layers in the `Qwen2.5-7B` model(**a, b** and **c**) and `Qwen2.5-7B-instruct` model(**d, e** and **f**).

appear to serve as critical junctions where structural perturbations most directly translate into downstream performance loss. Such robustness vulnerabilities likely stem from these layers' dual role in integrating lower-level token interactions and preparing higher-level semantic abstractions, making them both information-rich and sensitive to distributional shifts.

However, differences emerge when contrasting base versus chat- or instruction-tuned variants. In the base models (`Llama3-8B`, `Mistral-7B`; Figure ( 15, and 16) subfigures **a-c**), the correlation patterns outside the

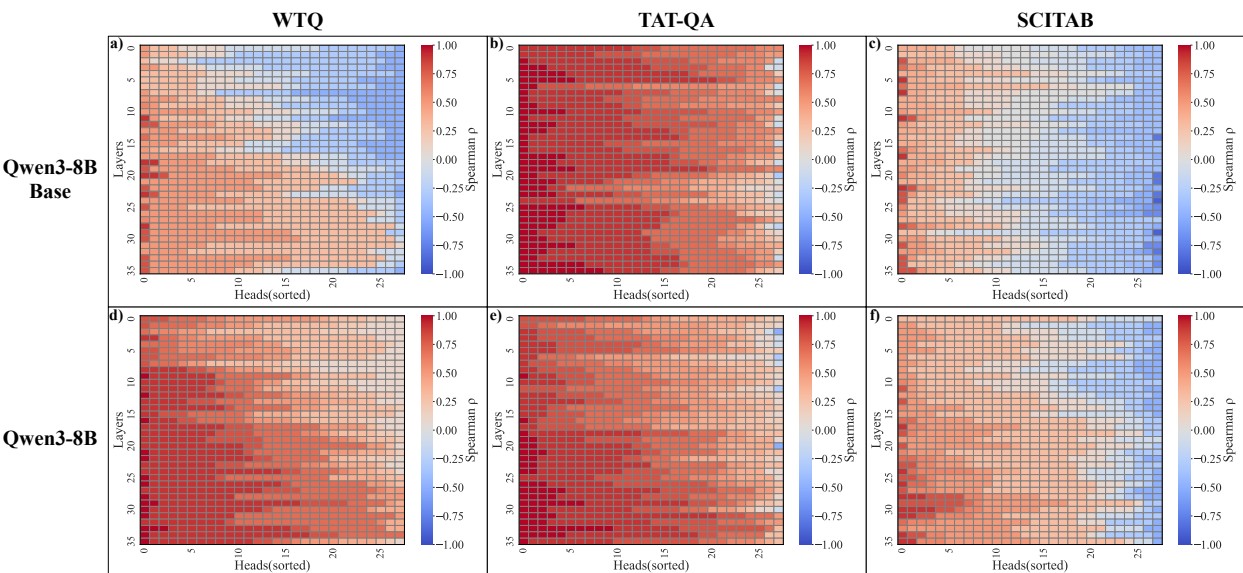

Figure 18: Heatmap showing Spearman correlation between changes in attention entropy and EM difference across all attention heads and layers in the `Qwen3-8B base` model(**a, b** and **c**) and `Qwen3-8B` model(**d, e** and **f**).

middle layers fluctuate markedly, with a mixture of weak or even negative correlation, especially apparent on **TAT-QA** and **SCITAB** tasks. Specifically, for Figure 16(**c**),`Mistral-7B` does not correlate because for all the questions EM was 0. These oscillations align with the base models' generally lower EM scores on these datasets, suggesting that noisy or unreliable predictions can obscure the coherent relationships between entropy and performance.

By contrast, the chat and instruct-adapted models (`Llama3-8B-Instruct`, `Mistral-7B-Instruct`; Figure (15 and 16) subfigures **d-f**) display stronger positive correlation, with some extending beyond the middle layers. On **WTQ** in particular, both the initial encoding layers (layers $0-2$) and the highest layers (uppermost $29-31$ blocks) contribute positively to the entropy–EM relation, suggesting that conversational and instruction tuning bolsters the model's resilience to perturbations at both the token-embedding stage and the final consolidation stage. This extended positive band likely reflects enhanced parameter alignment across the architecture, enabling more stable information propagation even under input disruptions.

In general, these results yield two primary implications. First, conversational and instruction-tuning systematically extends the alignment between attention entropy instabilities and performance degradation across a broader portion of the transformer hierarchy, thereby allowing perturbation robustness not only in the mid layers but also at its boundaries. Second, task domain complexity governs which layers most critically underpin model stability: general-domain benchmarks (e.g., **WTQ**) draw on a broad spectrum of transformer depths, whereas specialized datasets remain predominantly reliant on central layers.

Across the Qwen model families, the correlation heatmaps in Figures 17 and 18 reveal patterns broadly consistent with those reported for `Llama3` and `Mistral` in the main text. For both `Qwen 2.5` and `Qwen 3`, the middle transformer layers consistently exhibit the strongest positive Spearman correlations between perturbation-induced changes in attention entropy and EM degradation, reinforcing the interpretation of these layers as structural "choke points" where disruptions to schema alignment most directly translate into performance loss. Base variants (Figures 17a–c, 18a–c) show more fluctuation outside the mid-layer band, including weak or negative correlations on **TAT-QA** and **SCITAB**, mirroring the instability seen in `Llama3-8B` and `Mistral-7B`. This instability often coincides with lower overall EM, as in `Qwen 2.5-7B` on SCITAB. In contrast, the instruction-tuned counterparts (Figures 17d–f, 18d–f) display stronger and more spatially extended positive correlations, with **WTQ** in particular showing elevated values not only in the middle layers but also in the initial embedding layers (0–2) and uppermost layers, suggesting improved perturbation

robustness at both early encoding and final integration stages. These trends indicate that, for `Qwen` as for `Llama3` and `Mistral`, instruction tuning enhances the architectural alignment of attention–performance relationships, while the dataset domain continues to determine how localized or distributed the layerwise sensitivity appears.

