# OpenReview forum: "Exploring the Robustness of Language Models for Tabular Question Answering via Attention Analysis"
_TMLR — Accepted by TMLR_

### Review · Reviewer_JP73 · 2025-07-17

**Summary Of Contributions:**

The authors point out an interesting phenomenon: although large language models have already excelled in unstructured tasks, their structural reasoning mechanisms and robustness in Tabular QA (TQA) remain a “black box.” In real-world scenarios, row/column misalignment or numeric noise can distort answers, and existing work lacks systematic evaluations and explanations of internal attention dynamics. The authors design a benchmark covering row/column swaps, transposition, and various value perturbations, systematically evaluating the ICL robustness of LLMs of different scales and instruction tuning on WTQ, TAT-QA, and SCITAB, and quantifying performance degradation and sensitivity with Emd and VP. Through layer- and head-wise attention entropy analysis, they find that perturbation-induced entropy increases are highly correlated with EM drops and are most sensitive in the middle layers, and on this basis they call for developing structure-aware self-attention and domain-adaptive techniques.

**Audience:**

Yes

**Claims And Evidence:**

Yes

**Requested Changes:**

1.	Method level (critical)

    Add comparative experiments with structure-aware or models specifically trained for tabular tasks (e.g., TAPEX, TableFormer) (the authors may choose models that are easier to implement; the methods I suggest are only for reference and do not necessarily have to be compared with the ones I mention) to verify the generality of the paper’s findings and eliminate model-selection bias.

2.	Experimental metrics (improvement)

    In the evaluation, supplement semantic consistency metrics (such as F1 or Set EM) and report significance testing, to avoid underestimating the model’s true capability by relying solely on Exact Match.

3.	Table-size analysis (improvement)

    Extend experiments to larger or more complex tables and group-report performance and attention-entropy changes to examine whether the middle-layer sensitivity conclusion remains stable under different context-window lengths (my consideration here is that current large language models already show very strong understanding of short, small tables; for example, users can first convert a table into markdown format for better text comprehension by the model).

**Strengths And Weaknesses:**

Strengths
- The authors conducted comprehensive and rigorous robustness testing of multiple large language models, applying rich tabular perturbations—including structural transformations and value modifications—across diverse domains such as Wikipedia, financial reports, and scientific tables, and systematically revealed how in-context learning, model scale, and instruction tuning affect Tabular QA performance and stability.
- Secondly, through fine-grained analysis of model attention distributions, the paper finds that perturbation-induced attention dispersion is significantly correlated with performance degradation, and further points out that attention in the middle layers is most sensitive to table-structure changes, providing new insights into the models’ internal working mechanisms.

Weaknesses
- The study is limited to prompt-based Tabular QA with general-purpose LLMs and lacks comparison with models specially trained for table tasks or equipped with structure-aware optimizations, so it is unclear whether such specialized methods could mitigate the problems revealed or alter the conclusions.
- The paper uses attention weight changes as the main explanatory tool; prior work shows that attention distributions are not necessarily strictly aligned with true feature importance, therefore, evaluating model robustness solely by attention dispersion may be insufficient to uncover the deeper causes of errors.

---

> ### Author Response · Authors · 2025-07-29
> **Response to Reviewer JP73**
>
> We thank the reviewer for the feedback. Below are the detailed clarifications to the observations:
> ### Scope Beyond Prompt-Only
> Although our work aims to examine table comprehension ability, building on recent progress in the task of direct prompting [1, 2, 3, 4], we have extended our analysis to include structure-aware and table-specific models, namely TAPEX. We conducted the full perturbation suite on five TAPEX checkpoints (ranging from tiny to large).
> These models exhibit similar qualitative robustness trends to general-purpose LLMs: structural perturbations, such as column swaps and transpose operations, consistently lead to performance degradation. However, we find that all the variants of the models are more robust with column swap in comparison to other structural perturbations.
> Although TAPEX models achieve higher performance on WTQ (as evident from the provided performance tables), their generalization to other datasets (TAT-QA) is significantly weaker, highlighting a critical robustness-generalization tradeoff. Specifically, TAPEX performance sharply declines outside its training domain (WTQ), whereas general-purpose LLMs, despite their decent accuracy, consistently demonstrate stable performance across diverse domains without additional fine-tuning.
> Additionally, we clarify the rationale for not implementing it on the SCITAB task specifically for TAPEX. TAPEX is originally trained to solve synthetic SQL queries and is fine-tuned for table-based question answering and fact verification. Given that the SCITAB dataset can be formulated as a question-answering task, without fine-tuning on it, it is not possible to generate a scientific claims verification problem.
> Lastly, our earlier reference to TAPEX being "more constrained by the table" aims to emphasize that TAPEX heavily relies on exact table entries for accuracy, which is beneficial for faithful table comprehension. However, this also renders TAPEX vulnerable to structural perturbations, as evidenced by the significant drop in performance with different perturbations compared to LLMs, highlighting its narrower robustness compared to general LLMs.
> ### Attention Dispersion as a metric
> We appreciate the reviewer’s concern regarding the use of attention entropy and aligning with feature importance. However, we contend that attention entropy, used in our work, serves not as a definitive attribution method but as a principled metric for identifying structural fragility in LLM reasoning. We ground our choice in robust prior research which have shown that attention distributions reliably signal disruptions in contextual token alignment and reasoning [5–7]. More recently, Zhang et al. empirically demonstrated attention entropy as a proxy for diversity in context encoding [8]. Similarly, Zhai et al. and Araabi et al., respectively, showed that entropy stabilization and regularization directly correlate with performance stability and generalization improvements [9, 10].
> To directly address the reviewer’s concern, we emphasize that our analysis explicitly demonstrates a statistically significant correlation between attention entropy changes and performance degradation, validated not only on Llama-8B-Instruct but consistently across multiple models (Llama3 and Mistral variants) and datasets (WTQ, TAT-QA, SCITAB). This extensive cross-model validation strongly supports the robustness and generality of our findings.
> ### Additional Evaluation Metrics
> We thank the reviewer for suggesting additional comprehensive evaluation metrics. We have included F1 alongside EM to assess semantic consistency between model predictions and ground truth more finely. With F1, we observe more pronounced numerical differences compared to EM, but the performance trends remain consistent.
> Specifically, F1 allows us to capture partial semantic matches that EM cannot, highlighting cases where models produce partially correct answers under perturbation. Thus, F1 enriches our understanding of the models’ robustness by revealing nuanced behaviors. We have added the results with F1 separately.
> ### Table Size & Complexity Analysis
> We acknowledge the reviewer’s interest in understanding if our findings generalize to larger or more complex tables. In our original study, we carefully controlled for table size by restricting experiments to tables with fewer than 150 cells, isolating reasoning behavior from sequence-length constraints [11]. We have included a detailed comparison of EM scores across table sizes in Appendix E. Importantly, our complexity analysis explicitly demonstrates that structural perturbations disproportionately affect larger, more complex tables, causing greater performance degradation in these scenarios. This analysis directly links table complexity to increased sensitivity, reinforcing our primary findings regarding structural robustness and addressing the reviewer’s query.

---

> > ### Author Response · Authors · 2025-07-29
> > **References**
> >
> > References
> > [1] Chen, Wenhu. “Large Language Models Are Few(1)-Shot Table Reasoners.” Findings of the Association for Computational Linguistics: EACL 2023, 2023. Association for Computational Linguistics. https://aclanthology.org/2023.findings-eacl.83.pdf
> >
> > [2] Wang, Zilong, et al. “Chain-of-Table: Evolving Tables in the Reasoning Chain for Table Understanding.” International Conference on Learning Representations (ICLR 2024), 2024. https://openreview.net/forum?id=4L0xnS4GQM
> >
> > [3] Liu, Tianyang, Fei Wang, and Muhao Chen. “Rethinking Tabular Data Understanding with Large Language Models.” Proceedings of the 2024 Conference of the North American Chapter of the Association for Computational Linguistics (NAACL), 2024. Association for Computational Linguistics. https://arxiv.org/abs/2312.16702
> >
> > [4] Chen, Si-An, et al. “TableRAG: Million-Token Table Understanding with Language Models.” Advances in Neural Information Processing Systems (NeurIPS 2024), 2024. https://openreview.net/forum?id=41lovPOCo5
> >
> > [5] Zhao, Zheng, et al. ‘Layer by Layer: Uncovering Where Multi-Task Learning Happens in Instruction-Tuned Large Language Models’. Proceedings of the 2024 Conference on Empirical Methods in Natural Language Processing, Association for Computational Linguistics, 2024, pp. 15195–15214, https://doi.org/10.18653/v1/2024.emnlp-main.847.
> >
> > [6] Barbero, Federico, et al. ‘Why Do LLMs Attend to the First Token?’ arXiv [Cs], no. arXiv:2504.02732, arXiv, Apr. 2025, https://doi.org/10.48550/arXiv.2504.02732. arXiv.
> >
> > [7] Olsson, Catherine, et al. ‘In-Context Learning and Induction Heads’. arXiv [Cs], no. arXiv:2209.11895, arXiv, Sept. 2022, https://doi.org/10.48550/arXiv.2209.11895. arXiv.
> >
> > [8] Zhang et al. 2024. Attention Entropy is a Key Factor: An Analysis of Parallel Context Encoding with Full-attention-based Pre-trained Language Models.
> >
> > [9] Zhai et al., 2023. Stabilizing Transformer Training by Preventing Attention Entropy Collapse.
> >
> > [10] Araabi et al., 2024. Entropy– and Distance-Regularized Attention Improves Low-Resource Neural Machine Translation.
> >
> > [11] Modarressi, Ali, et al. NoLiMa: Long-Context Evaluation Beyond Literal Matching. Forty-Second International Conference on Machine Learning, 2025, https://openreview.net/forum?id=0OshX1hiSa.

---

> > ### Author Response · Authors · 2025-07-29
> > **F1 score for TAPAS model**
> >
> > # F1 score for TAPAS model
> >
> > ### WTQ_DATA
> > |Model|Original|Column Swap|Row Swap|Transpose|Transpose Row Swap|Transpose Col Swap|DVP|RVP|NVP|
> > |-----|--------|-----------|--------|---------|------------------|------------------|---|---|---|
> > TAPAS Tiny|0.584|0.525|0.282|0.094|0.079|0.091|0.390|0.451|0.064|
> > TAPAS Small|0.717|0.705|0.422|0.087|0.098|0.061|0.665|0.621|0.069|
> > TAPAS Mini|0.717|0.683|0.423|0.120|0.087|0.074|0.635|0.633|0.082|
> > TAPAS Medium|0.737|0.704|0.498|0.085|0.062|0.078|0.712|0.643|0.064|
> > TAPAS Base|0.720|0.710|0.484|0.097|0.060|0.105|0.706|0.624|0.076|
> > TAPAS Large|0.735|0.716|0.519|0.120|0.096|0.109|0.709|0.611|0.089|
> >
> > ### TATQA_DATA
> > |Model|Original|Column Swap|Row Swap|Transpose|Transpose Row Swap|Transpose Col Swap|
> > |-----|--------|-----------|--------|---------|------------------|------------------|
> > TAPAS Tiny|0.031|0.026|0.026|0.047|0.035|0.046|
> > TAPAS Small|0.050|0.043|0.031|0.100|0.060|0.099|
> > TAPAS Mini|0.053|0.052|0.024|0.112|0.062|0.103|
> > TAPAS Medium|0.057|0.061|0.032|0.145|0.084|0.144|
> > TAPAS Base|0.052|0.049|0.033|0.083|0.063|0.081|
> > TAPAS Large|0.045|0.052|0.039|0.106|0.074|0.115|

---

> > ### Author Response · Authors · 2025-07-29
> > **F1 score for all evaluated models**
> >
> > # F1 score for all evaluated models
> > Note: The labels **B**, **C**, and **I** refers to the Base, Chat and Instruct variant of the model.
> > ## WTQ_DATA
> > |Model|Original|Column Swap|Row Swap|Transpose|Transpose Row Swap|Transpose Col Swap|NT|DVP|RVP|NVP|
> > |-----|--------|-----------|--------|---------|------------------|------------------|--|---|---|---|
> > Llama2 7B|0.173|0.169|0.134|0.127|0.129|0.090|0.032|0.053|0.022|0.028|
> > Llama3 8B|0.316|0.254|0.252|0.259|0.212|0.193|0.027|0.187|0.112|0.027|
> > Mistral 7B|0.322|0.321|0.252|0.262|0.243|0.214|0.037|0.139|0.083|0.036|
> > Qwen2.5  7B|0.271|0.254|0.206|0.206|0.212|0.161|0.042|0.111|0.044|0.040|
> > Qwen3  8B **B**|0.350|0.302|0.252|0.318|0.286|0.191|0.043|0.173|0.091|0.039|
> > Llama2 70B|0.308|0.331|0.292|0.268|0.246|0.200|0.038|0.150|0.079|0.054|
> > Llama3 70B|0.414|0.400|0.354|0.380|0.352|0.277|0.105|0.217|0.139|0.109|
> > Mistral 8x7B|0.305|0.313|0.258|0.291|0.275|0.239|0.040|0.162|0.082|0.061|
> > Qwen2.5  72B|0.432|0.426|0.364|0.415|0.431|0.270|0.037|0.250|0.159|0.069|
> > Qwen3  30B **B**|0.313|0.272|0.275|0.299|0.247|0.202|0.045|0.168|0.108|0.059|
> > ||||||||||||
> > Llama2 7B **C**|0.269|0.271|0.204|0.209|0.171|0.145|0.057|0.146|0.078|0.040|
> > Llama3 8B **I**|0.460|0.434|0.410|0.386|0.368|0.292|0.022|0.336|0.235|0.072|
> > Mistral 7B **I**|0.417|0.409|0.343|0.371|0.351|0.257|0.025|0.327|0.289|0.065|
> > Qwen2.5  7B **I**|0.459|0.442|0.397|0.415|0.366|0.306|0.026|0.335|0.233|0.090|
> > Qwen3  8B|0.347|0.302|0.252|0.318|0.286|0.191|0.032|0.173|0.091|0.039|
> > Llama2 70B **C**|0.339|0.341|0.282|0.272|0.281|0.215|0.024|0.214|0.134|0.056|
> > Llama3 70B **I**|0.619|0.605|0.531|0.611|0.574|0.466|0.109|0.491|0.383|0.130|
> > Mistral 8x7B **I**|0.406|0.380|0.319|0.386|0.334|0.253|0.057|0.203|0.113|0.072|
> > Qwen2.5  72B **I**|0.589|0.590|0.511|0.551|0.563|0.432|0.091|0.475|0.265|0.103|
> > Qwen3  30B|0.368|0.345|0.301|0.340|0.304|0.245|0.041|0.240|0.187|0.064|
> >
> >
> > ## TATQA_DATA
> > |Model|Original|Column Swap|Row Swap|Transpose|Transpose Row Swap|Transpose Col Swap|NT|
> > |-----|--------|-----------|--------|---------|------------------|------------------|--|
> > Llama2 7B|0.216|0.191|0.185|0.179|0.146|0.163|0.014|
> > Llama3 8B|0.348|0.294|0.306|0.326|0.264|0.297|0.015|
> > Mistral 7B|0.329|0.292|0.306|0.295|0.235|0.272|0.021|
> > Qwen2.5  7B|0.282|0.254|0.251|0.270|0.233|0.264|0.020|
> > Qwen3  8B **B**|0.304|0.280|0.270|0.285|0.256|0.266|0.022|
> > Llama2 70B|0.378|0.349|0.340|0.330|0.283|0.328|0.013|
> > Llama3 70B|0.450|0.423|0.424|0.421|0.389|0.403|0.054|
> > Mistral 8x7B|0.364|0.322|0.319|0.333|0.256|0.325|0.031|
> > Qwen2.5  72B|0.388|0.358|0.361|0.365|0.327|0.364|0.020|
> > Qwen3  30B **B**|0.384|0.351|0.355|0.371|0.290|0.328|0.031|
> > |||||||||
> > Llama2 7B **C**|0.230|0.198|0.210|0.199|0.156|0.175|0.028|
> > Llama3 8B **I**|0.350|0.305|0.321|0.340|0.279|0.311|0.005|
> > Mistral 7B **I**|0.312|0.261|0.272|0.286|0.240|0.267|0.024|
> > Qwen2.5  7B **I**|0.306|0.285|0.281|0.294|0.260|0.284|0.018|
> > Qwen3  8B|0.316|0.301|0.321|0.320|0.279|0.287|0.033|
> > Llama2 70B **C**|0.307|0.268|0.282|0.270|0.221|0.248|0.004|
> > Llama3 70B **I**|0.523|0.499|0.486|0.514|0.452|0.472|0.054|
> > Mistral 8x7B **I**|0.372|0.336|0.327|0.326|0.269|0.325|0.036|
> > Qwen2.5  72B **I**|0.426|0.426|0.411|0.418|0.377|0.400|0.040|
> > Qwen3  30B|0.409|0.373|0.373|0.371|0.319|0.329|0.034|
> >
> > ## SCITAB_DATA
> >
> > |Model|Original|Column Swap|Row Swap|Transpose|Transpose Row Swap|Transpose Col Swap|NT|
> > |-----|--------|-----------|--------|---------|------------------|------------------|--|
> > Llama2 7B|0.001|0.003|0.002|0.002|0.002|0.001|0.000|
> > Llama3 8B|0.341|0.351|0.323|0.345|0.354|0.320|0.000|
> > Mistral 7B|0.000|0.000|0.000|0.000|0.000|0.000|0.000|
> > Qwen2.5  7B|0.003|0.001|0.002|0.003|0.003|0.002|0.000|
> > Qwen3  8B **B**|0.329|0.297|0.304|0.318|0.284|0.283|0.000|
> > Llama2 70B|0.034|0.027|0.023|0.042|0.036|0.032|0.000|
> > Llama3 70B|0.380|0.366|0.352|0.369|0.365|0.356|0.000|
> > Mistral 8x7B|0.000|0.000|0.000|0.000|0.000|0.000|0.000|
> > Qwen2.5  72B|0.353|0.305|0.295|0.344|0.315|0.365|0.000|
> > Qwen3  30B **B**|0.105|0.074|0.085|0.094|0.086|0.067|0.000|
> > |||||||||
> > Llama2 7B **C**|0.384|0.387|0.383|0.378|0.385|0.387|0.000|
> > Llama3 8B **I**|0.328|0.322|0.331|0.322|0.333|0.320|0.000|
> > Mistral 7B **I**|0.290|0.286|0.287|0.285|0.284|0.288|0.000|
> > Qwen2.5  7B **I**|0.371|0.335|0.369|0.376|0.359|0.351|0.000|
> > Qwen3  8B|0.383|0.348|0.332|0.383|0.364|0.330|0.000|
> > Llama2 70B **C**|0.084|0.072|0.076|0.091|0.077|0.078|0.000|
> > Llama3 70B **I**|0.478|0.465|0.460|0.466|0.458|0.454|0.000|
> > Mistral 8x7B **I**|0.083|0.080|0.080|0.084|0.087|0.083|0.000|
> > Qwen2.5  72B **I**|0.632|0.585|0.585|0.628|0.579|0.603|0.000|
> > Qwen3  30B|0.038|0.030|0.021|0.022|0.024|0.015|0.000|

---

### Review · Reviewer_sAwL · 2025-07-19

**Summary Of Contributions:**

It is an analysis paper measuring the relation of attention weights and tabular question answering capacity of large language models (LLMs). Basic idea is to measure the dispersion of attention pattern when looking at tabular information in order to retrieve information by differentiating the difficulties of tables and perturbation.  Experiments show that the attention pattern shifts, measured by entropy, are correlated with degradation in performance on three datasets coming from Wikipedia, financial domain and scientific domain.

**Audience:**

No

**Claims And Evidence:**

No

**Requested Changes:**

* This work needs discussion on why focusing on tabular question answering task is suitable to measure the changes in attention patterns and performance. For example, it is possible to run comparison with other tasks, e.g., QAs with long context [1].
* Further discussion regarding the use of entropy metric will be necessary to justify the use of the metrics with related studies [1, 2, 3].
* This work needs detail explanation for the three tasks settings, e.g, what kind of challenges shared or specific to each dataset.
* Experiments should be controlled by several factors, e.g., size of a table, information in each cell, that might affect the difficulties of the tabular understanding.
* Further discussion and/or experiment is necessary to separate the knowledge memorization impact to see whether an LLM can answer without looking at any tabular information. For example, it is possible to systematically randomize the labels in order to completely remove the impact of knowledge [4], not partly adding noises.
* It is not what is plotted in Figure 1 (b), Figure 6 (b) and (c), whether they are showing average of all the information or not.

References

[1] Zhang et al. 2024. Attention Entropy is a Key Factor: An Analysis of Parallel Context Encoding with Full-attention-based Pre-trained Language Models.

[2] Zhai et al., 2023. Stabilizing Transformer Training by Preventing Attention Entropy Collapse.

[3] Araabi et al., 2024. Entropy– and Distance-Regularized Attention Improves Low-Resource Neural Machine Translation.

[4] Sakai et al., 2024. Does Pre-trained Language Model Actually Infer Unseen Links in Knowledge Graph Completion?

**Strengths And Weaknesses:**

Strengths
* Analysis on the tabular question answering datasets is systematically conducted by measuring the relation of perturbation of the tabular information, performance differences and changes in attention entropy.

Weaknesses
* This work split the experiments by three datasets, but it is not clear the impact of difficulties of the tasks. First, no discussion exists regarding the characteristics of the three datasets. Second, this work needs further split by the size of table and information in each cell, e.g., numeric values or textual information, for a more comprehensive studies to measure the impact of difficulties.
* The impact of knowledge memorized in LLMs is not clear. Given the drop of performance when adding noise to cell, LLMs might heavily rely on the memorized knowledge and this might have an impact to the attention pattern.
* Several clarity issues exist in this work. For example, the scatter plot in Figure 1 (b) is not clear what is plotted, whether it is showing only averages of layers and heads.

---

> ### Author Response · Authors · 2025-07-29
> **Response to Reviewer sAwL**
>
> We appreciate the reviewer’s thoughtful feedback and provide detailed responses to the comments below.
> ### Dataset Characterization and Task Difficulty
> We selected these three tabular datasets — WTQ (Wikipedia tables), TAT-QA (financial tables), and SCITAB (scientific tables) — for their domain diversity and semantic/task complexity. WTQ emphasizes general knowledge content; TAT-QA introduces reasoning over textual and financial entries; and SCITAB combines compositional logic with scientific tabular facts. WTQ primarily includes a mixture of textual and numeric data, emphasizing general knowledge and multi-step reasoning. TAT-QA predominantly contains numeric financial data combined with textual descriptions, requiring numerical reasoning and text-table integration. In contrast, SCITAB primarily involves textual scientific claims with numeric data supporting structured logical reasoning.
>
> As you mentioned, the table size-based control, we explicitly controlled for table size (<150 cells) in our experiments (Appendix C) to avoid confounding due to length-based degradation and focused solely on table comprehension[1]. Furthermore, Appendix E and Figure 13 provide a detailed breakdown of performance across table size bins, showcasing how models behave under increasing table complexity. Although we briefly describe the different properties of datasets in Appendices B, C, and E, we will include a dedicated dataset section in the appendix to provide a more comprehensive description of the dataset.
>
> ### LLM Memorization
> This is a key concern we directly address via value perturbation experiments (Section 2.2, Table 1).
> The No Table (NT) and Null Value Perturbation (NVP) settings serve as direct probes for memorization. Performance on WTQ under NT (up to 6%) and NVP (7%) drops dramatically compared to the original inputs (39%), as shown in Table 1. This suggests that some questions are answered via memorized facts, as there are no answers within the provided tabular context; LLM could still answer the question. Given this in mind, we extend our analysis to other datasets (TAT-QA and SCITAB), which are non-open-domain datasets and are less likely to appear in pretraining corpora, showing near-zero NT performance.
> We are thankful that the reviewer mentioned Chen et al.’s work in our preview. Our value-based perturbation strategy shares conceptual similarities with the framework proposed by Chen et al. in their study of knowledge memorization in pre-trained language models. Specifically, our DVP aligns with their Virtual World setup, where semantically plausible but factually incorrect answers are introduced while preserving entity types and syntactic structure. Similarly, our RVP  mirrors their Anonymized Entity setting, where arbitrary or nonsensical tokens are substituted to break entity-level associations. Additionally, our NVP and NT scenarios systematically remove access to the correct answer, akin to their effort to disentangle surface-level lexical memorization from genuine relational inference. Across these setups, we observe substantial performance degradation, providing strong empirical evidence that LLMs are sensitive to value-level disruptions even when structural information is intact.
> Similarly, for attention analysis, all structural perturbations (e.g., transpose, row swap) operate under a constant memorization prior, allowing us to compare robustness in a controlled way. Our attention analysis thus measures changes in LLM internal behavior in response to structural variation, rather than raw knowledge access; this would be difficult to control as an experimental design, so we measure the dispersion of attention for the structural augmentation.
> Although we did not try randomization of labels due to practical constraints on maintaining semantic consistency, our perturbation methods sufficiently approximate and highlight the memorization effect [2]. We will explicitly state this reasoning in the revised manuscript and consider systematic randomization for future studies.

---

> > ### Author Response · Authors · 2025-07-29
> > **Additional Comment**
> >
> > ### Task Selection Justification
> > Tabular QA (TQA) offers unique advantages for studying structural sensitivity. Unlike general QA or long-context reading comprehension, TQA requires spatial reasoning and schema interpretation, e.g., understanding row-column alignments. Structural perturbations, such as row/column swaps, provide fine-grained control to probe model vulnerabilities in attention alignment. We acknowledge the potential comparison with long-context QA tasks (Modarressi et al., 2025) [1]. However, TQA uniquely provides structured perturbations that precisely target spatial reasoning without introducing confounding text length variables. This makes TQA particularly suitable for analyzing attention pattern changes explicitly due to structural perturbations.
> >
> > ### Metric Justification and Clarity
> > We agree that attention entropy must be well-motivated. In our paper, we cite and build on prior works, which analyze attention dispersion phenomena [3,4,5]. Explicitly addressing the reviewer’s concern, attention entropy is particularly suited for tables due to its sensitivity to structured disruptions. Perturbations such as row or column swaps inherently disrupt the alignment patterns critical for table comprehension, leading to measurable entropy shifts. Thus, entropy is uniquely valuable for capturing model sensitivity specifically to tabular structural perturbations, distinct from general context encoding.
> > Our paper finds a strong positive correlation between perturbation-induced entropy changes and performance degradation, particularly in the middle layers, aligning with Zhang et al., who emphasize the role of entropy in parallel context encoding by empirically demonstrating that entropy serves as a proxy for attention diversity, crucial for robust language understanding [6]. Similarly, Zhai et al. show that stabilizing entropy prevents attention collapse during training, while Araabi et al. demonstrate that entropy regularization improves generalization in low-resource settings [7,8]. By extending these ideas to structured data, the paper leverages entropy not only as an interpretability tool but also as a principled proxy for identifying attention instability under perturbation, thereby supporting its broader applicability in evaluating model robustness. While studies like Zhang et al. explore entropy in parallel context encoding, our work uniquely applies entropy to structured perturbations in tables, a domain where attention misalignment is especially consequential to the end performance [6].
> >
> > ### On Clarity of Figures (e.g., Figure 1(b), 6(b), 6(c))
> > Thank you for highlighting this. Figure 1(b) plots the average attention entropy change against EM scores across perturbation types for Llama-8B-Instruct on WTQ, averaged over all attention heads and data points, resulting in one data point per perturbation type. Figures 6(b) and 6(c), however, represent individual attention head-layer pairs (e.g., Layer 0 Head 10, Layer 19 Head 23), where each point averages only across data points for each specific head-layer pair, explicitly highlighting head-specific variability in sensitivity. We will clarify this distinction explicitly in figure captions and text. Thank you again for the insightful comments; we have incorporated these explicit clarifications and adjustments in our revised manuscript.
> >
> >
> >
> > References
> >
> > [1] Modarressi, Ali, et al. NoLiMa: Long-Context Evaluation Beyond Literal Matching. Forty-Second International Conference on Machine Learning, 2025, https://openreview.net/forum?id=0OshX1hiSa.
> >
> > [2] Sakai et al., 2024. Does Pre-trained Language Model Actually Infer Unseen Links in Knowledge Graph Completion?
> >
> > [3] Zhao, Zheng, et al. ‘Layer by Layer: Uncovering Where Multi-Task Learning Happens in Instruction-Tuned Large Language Models’. Proceedings of the 2024 Conference on Empirical Methods in Natural Language Processing, Association for Computational Linguistics, 2024, pp. 15195–15214, https://doi.org/10.18653/v1/2024.emnlp-main.847.
> >
> > [4] Barbero, Federico, et al. ‘Why Do LLMs Attend to the First Token?’ arXiv [Cs], no. arXiv:2504.02732, arXiv, Apr. 2025, https://doi.org/10.48550/arXiv.2504.02732. arXiv.
> >
> > [5] Olsson, Catherine, et al. ‘In-Context Learning and Induction Heads’. arXiv [Cs], no. arXiv:2209.11895, arXiv, Sept. 2022, https://doi.org/10.48550/arXiv.2209.11895. arXiv.
> >
> > [6] Zhang et al. 2024. Attention Entropy is a Key Factor: An Analysis of Parallel Context Encoding with Full-attention-based Pre-trained Language Models.
> >
> > [7] Zhai et al., 2023. Stabilizing Transformer Training by Preventing Attention Entropy Collapse.
> >
> > [8] Araabi et al., 2024. Entropy– and Distance-Regularized Attention Improves Low-Resource Neural Machine Translation.

---

### Review · Reviewer_gQzM · 2025-07-20

**Summary Of Contributions:**

The paper presents an analysis of how LLMs' performance on "structured comprehension tasks" depends on how information is presented -- essentially, what makes it easier or harder for LLMs to extract information from tables. Authors create "perturbed" variants of three test sets (WTQ, TAT-QA, SCITAB) by (1) rearranging the columns and rows of the table (structural perturbation, SP) and (2) applying value perturbation (ValP) in 4 different ways: changing the actual value while preserving the data type, using a random value, removing the correct answer, and not providing a table at all. Authors also provide an analysis of attention entropy in internal layers and show how it correlates with task performance.

**Audience:**

Yes

**Claims And Evidence:**

No

**Requested Changes:**

Please provide results with more recent models.

Please provide a better presentation of results so that readers can follow the main argument. E.g. in Table 1, maybe provide averages of the different EM/VP/EMD "scores" across the different perturbations, so that a "big picture" can emerge. Similarly, in Figures 1 and 6, provide an aggregate analysis that shows that there is some correlation between attention entropy and task performance beyond individual heads.

**Strengths And Weaknesses:**

Strengths

Understanding biases and preferences in LLMs is an important topic, empirical studies of that data are needed
The paper is thorough in its core approach, authors document their work and experiments
The paper is generally well written and the overall argument is intuitive

Weaknesses

The paper overstates its claims, such as "newer architectures like Llama3 are more
effective at table reasoning tasks" -- authors only evaluate Llama3 and Mistral models (incl instruction-tuned variants), and no more recent "architecture" (or training corpus) has been tested
I find the tasks, especially "value perturbation," ill-posed: one should not use an LLM to explicitly parse a table (but of course it can be done) -- because the LLM will mingle its internalized knowledge with the table information.
The paper is purely empirical and results are likely to be different for every new LLM, every new data type, etc -- so very little generalizable results are found.
The study on attention entropy is interesting but mostly descriptive.
I find it difficult to understand the main findings of the paper. For example, I cannot follow "Table 1 also distinctly indicates that LLMs that have undergone instruction or conversation-based fine-tuning outperform their base counterparts in TQA tasks for SCITAB dataset." -- Table 1 is huge and I don't see instruction-tuned models mentioned there.

---

> ### Author Response · Authors · 2025-07-29
> **Response to Reviewer gQzM**
>
> We thank the reviewer for providing their excellent insight into our work.
>
> ### Overstated Claims on LLM Architecture Performance:
> We agree that a broader architectural comparison would enhance generalizability. However, our claim is specifically bound to open weights and the popular models like Qwen, Llama, and Mistral that have been utilized for direct prompting of LLM for TQA tasks. I understand the oversight in the statement, “*newer architectures … reasoning tasks.*” We will modify it to reflect that it only addresses the variant of the models based on their family (“Llama”, “Mistral”, and “Qwen model”). We will also incorporate the evaluation of the Llama2 models to facilitate a direct comparison with the Llama3 and Qwen2.5 models, as well as the Qwen3 model. The new models we implemented are Llama2(7B and 70B), Llama2-chat(7B and 70B),Qwen2.5(7B and 72B), Qwen2.5-Instruct(7B and 72B), Qwen3 Base(8B and 30B) and Qwen3(8B and 30B). Similarly, we also implemented the TAPEX model, as suggested by Reviewer JP73, with five sizes: Tiny, Small, Mini, Medium, and Large, all of which were fine-tuned on WTQ datasets. The results (table below) suggest that the Llama3 and Qwen3 models are indeed more effective at the TQA task. The table below lists the average EM score of all the models over three different iterations:
>
>
> ### Concerns Over Value Perturbation Design
> We anticipate and address this confound in multiple ways. The main objective of value-based perturbation is to reflect how much of the knowledge is derived from the result of memorized knowledge, rather than being constrained by the provided prompt.  The “No Table” (NT) condition explicitly tests reliance on prior knowledge by removing the table entirely. As shown in Table 1, performance drops to zero(Table 1 and Figure 2) for non-WTQ datasets under NT, unlike the WTQ dataset, where model performance is more than 1% for NT, confirming that models rely heavily on the table content provided within the context in these domains. The Null Value Perturbation (NVP) and Random Value Perturbation (RVP) conditions further isolate the table signal from memorized knowledge by altering or erasing the correct cell content. We will make this reasoning more explicit and highlight NT, NVP, and RVP comparisons as key controls for testing memorization.
>
> ### Presentation and Clarity Issues (Table 1, Figures 1 & 6)
> To enhance the clarity and interpretability of our results, we will revise Table 1 to present more distinct results. This will highlight key trends across perturbations and models, allowing readers to better understand the overall impact of different perturbation types and model characteristics on performance.
> We do want to note that we had an oversight in mentioning, "Table 1 also distinctly indicates that LLMs that have undergone instruction or conversation-based fine-tuning … for SCITAB dataset." We meant to say Figure 2, where it includes the average EM for the individual model over different perturbations. And here we see that, in general, instruction- or conversation-based fine-tuning outperforms its base counterparts for all models.
> We would like to clarify that Figure 1 displays a scatter plot illustrating the relationship between changes in attention entropy and EM scores across different structural perturbation types, with values averaged over all data points and attention heads—each point representing a specific perturbation. In contrast, Figures 6(b) and 6(c) present scatter plots of entropy-EM correlations at the granularity of individual attention heads, with each point reflecting the average across data points for a specific head-layer pair, highlighting the variability in sensitivity among heads. In our revision, we will update the figure captions and corresponding text to distinguish between averaged results and head-specific analyses clearly, ensuring readers can readily interpret the scope and granularity of each visualization.
>
> References
>
> [1] Chen, Wenhu. “Large Language Models Are Few(1)-Shot Table Reasoners.” Findings of the Association for Computational Linguistics: EACL 2023, 2023. Association for Computational Linguistics. https://aclanthology.org/2023.findings-eacl.83.pdf
>
> [2] Wang, Zilong, et al. “Chain-of-Table: Evolving Tables in the Reasoning Chain for Table Understanding.” International Conference on Learning Representations (ICLR 2024), 2024. https://openreview.net/forum?id=4L0xnS4GQM
>
> [3] Liu, Tianyang, Fei Wang, and Muhao Chen. “Rethinking Tabular Data Understanding with Large Language Models.” Proceedings of the 2024 Conference of the North American Chapter of the Association for Computational Linguistics (NAACL), 2024. Association for Computational Linguistics. https://arxiv.org/abs/2312.16702
>
> [4] Chen, Si-An, et al. “TableRAG: Million-Token Table Understanding with Language Models.” Advances in Neural Information Processing Systems (NeurIPS 2024), 2024. https://openreview.net/forum?id=41lovPOCo5

---

> > ### Author Response · Authors · 2025-07-29
> > **EM score for all newly evaluated model**
> >
> > # EM score for models from Llama2, Qwen2.5, and Qwen3 variants with TAPEX model.
> > Note: The labels **B**, **C**, and **I** refers to the Base, Chat and Instruct variant of the model.
> > ## WTQ_DATA
> > |Model|Original|Column Swap|Row Swap|Transpose|Transpose Row Swap|Transpose Col Swap|NT|DVP|RVP|NVP|
> > |-----|--------|-----------|--------|---------|------------------|------------------|--|---|---|---|
> > Llama2 7B|0.150|0.148|0.118|0.110|0.108|0.078|0.021|0.033|0.012|0.014|
> > Qwen2.5  7B|0.191|0.196|0.155|0.158|0.160|0.131|0.023|0.064|0.009|0.026|
> > Qwen3  8B **B**|0.297|0.246|0.187|0.272|0.239|0.168|0.026|0.139|0.057|0.031|
> > Llama2 70B|0.273|0.299|0.260|0.235|0.229|0.178|0.044|0.135|0.061|0.038|
> > Qwen2.5  72B|0.370|0.369|0.306|0.366|0.375|0.235|0.020|0.227|0.137|0.043|
> > Qwen3  30B **B**|0.265|0.237|0.225|0.252|0.209|0.163|0.015|0.130|0.076|0.035|
> > ||||||||||||
> > Llama2 7B **C**|0.225|0.233|0.176|0.181|0.142|0.123|0.029|0.121|0.064|0.011|
> > Qwen2.5  7B **I**|0.335|0.312|0.292|0.273|0.242|0.190|0.002|0.236|0.158|0.057|
> > Qwen3  8B|0.295|0.246|0.187|0.272|0.239|0.168|0.010|0.139|0.057|0.031|
> > Llama2 70B **C**|0.306|0.302|0.248|0.250|0.253|0.201|0.030|0.187|0.109|0.045|
> > Qwen2.5  72B **I**|0.499|0.501|0.422|0.472|0.487|0.365|0.062|0.369|0.191|0.061|
> > Qwen3  30B|0.261|0.225|0.204|0.240|0.209|0.173|0.018|0.137|0.083|0.038|
> >
> >
> > ## TATQA_DATA
> > |Model|Original|Column Swap|Row Swap|Transpose|Transpose Row Swap|Transpose Col Swap|NT|
> > |-----|--------|-----------|--------|---------|------------------|------------------|--|
> > Llama2 7B|0.184|0.160|0.154|0.147|0.119|0.133|0.004|
> > Qwen2.5  7B|0.226|0.195|0.197|0.216|0.181|0.208|0.003|
> > Qwen3  8B **B**|0.254|0.229|0.218|0.236|0.208|0.216|0.000|
> > Llama2 70B|0.314|0.283|0.275|0.263|0.222|0.267|0.005|
> > Qwen2.5  72B|0.330|0.300|0.306|0.305|0.268|0.307|0.003|
> > Qwen3  30B **B**|0.336|0.302|0.304|0.320|0.239|0.280|0.006|
> > |||||||||
> > Llama2 7B **C**|0.185|0.159|0.168|0.155|0.124|0.135|0.008|
> > Qwen2.5  7B **I**|0.240|0.222|0.219|0.227|0.197|0.218|0.000|
> > Qwen3  8B|0.253|0.241|0.257|0.249|0.216|0.226|0.007|
> > Llama2 70B **C**|0.250|0.211|0.226|0.211|0.172|0.197|0.002|
> > Qwen2.5  72B **I**|0.353|0.353|0.337|0.340|0.303|0.330|0.008|
> > Qwen3  30B|0.346|0.305|0.304|0.298|0.258|0.260|0.010|
> >
> >
> > ## SCITAB_DATA
> > |Model|Original|Column Swap|Row Swap|Transpose|Transpose Row Swap|Transpose Col Swap|NT|
> > |-----|--------|-----------|--------|---------|------------------|------------------|--|
> > Llama2 7B|0.001|0.003|0.002|0.002|0.002|0.001|0.000|
> > Qwen2.5  7B|0.002|0.000|0.002|0.002|0.003|0.002|0.000|
> > Qwen3  8B **B**|0.327|0.293|0.301|0.312|0.280|0.278|0.000|
> > Llama2 70B|0.034|0.027|0.022|0.040|0.036|0.031|0.000|
> > Qwen2.5  72B|0.349|0.300|0.293|0.341|0.311|0.359|0.000|
> > Qwen3  30B **B**|0.093|0.065|0.074|0.082|0.078|0.059|0.000|
> > |||||||||
> > Llama2 7B **C**|0.384|0.387|0.382|0.378|0.383|0.387|0.000|
> > Qwen2.5  7B **I**|0.352|0.322|0.353|0.364|0.347|0.339|0.000|
> > Qwen3  8B|0.378|0.341|0.325|0.378|0.356|0.322|0.000|
> > Llama2 70B **C**|0.084|0.072|0.076|0.091|0.077|0.078|0.000|
> > Qwen2.5  72B **I**|0.630|0.578|0.579|0.626|0.575|0.600|0.000|
> > Qwen3  30B|0.021|0.022|0.012|0.008|0.015|0.007|0.000|
> >
> >
> > # TAPAS Model
> > ## WTQ_DATA
> > |Model|Original|Column Swap|Row Swap|Transpose|Transpose Row Swap|Transpose Col Swap|DVP|RVP|NVP|
> > |-----|--------|-----------|--------|---------|------------------|------------------|---|---|---|
> > TAPAS Tiny|0.562|0.507|0.230|0.078|0.064|0.078|0.355|0.418|0.064|
> > TAPAS Small|0.700|0.691|0.389|0.078|0.083|0.049|0.645|0.603|0.064|
> > TAPAS Mini|0.700|0.666|0.398|0.113|0.078|0.059|0.596|0.610|0.071|
> > TAPAS Medium|0.720|0.686|0.478|0.074|0.059|0.069|0.695|0.624|0.057|
> > TAPAS Base|0.705|0.696|0.465|0.083|0.044|0.088|0.695|0.603|0.050|
> > TAPAS Large|0.713|0.700|0.485|0.116|0.096|0.103|0.702|0.596|0.057|
> >
> > ## TATQA_DATA
> > |Model|Original|Column Swap|Row Swap|Transpose|Transpose Row Swap|Transpose Col Swap|
> > |-----|--------|-----------|--------|---------|------------------|------------------|
> > TAPAS Tiny|0.017|0.012|0.010|0.040|0.025|0.035|
> > TAPAS Small|0.032|0.030|0.022|0.093|0.052|0.092|
> > TAPAS Mini|0.045|0.046|0.015|0.105|0.057|0.097|
> > TAPAS Medium|0.046|0.046|0.019|0.125|0.075|0.132|
> > TAPAS Base|0.039|0.040|0.021|0.071|0.054|0.065|
> > TAPAS Large|0.037|0.045|0.029|0.085|0.058|0.087|

---

> > > ### Author Response · Authors · 2025-08-07
> > > **Attention Analysis with Qwen Model**
> > >
> > > We will also redraw Figure 7 to include the analysis with Attention Entropy with additional models. The result below includes the tabular format of Figure 7 with the new Spearman correlation of change in attention entropy to change in performance.
> > > ## WTQ dataset
> > > | Model | Spearman Correlation|
> > > |-------|---------------------|
> > > | Llama2-7b | 0.000000|
> > > | Llama2-7b **C**  | 0.700000|
> > > | Llama3-8b | 0.900000|
> > > | Llama3-8b **I**  | 0.900000|
> > > | Mistral-7b | 0.700000|
> > > | Mistral-7b **I**  | 0.700000|
> > > | Qwen2.5-7b | 0.600000|
> > > | Qwen2.5-7b **I**  | 0.600000|
> > > | Qwen3-8b | 0.900000|
> > > | Qwen3-8b **B**  | 0.500000|
> > >
> > > ## TATQA Dataset
> > > | Model | Spearman Correlation|
> > > |-------|---------------------|
> > > | Llama2-7b | 0.500000|
> > > | Llama2-7b **C**  | 0.872082|
> > > | Llama3-8b | 0.223607|
> > > | Llama3-8b **I**  | 0.800000|
> > > | Mistral-7b | 0.700000|
> > > | Mistral-7b **I**  | 0.400000|
> > > | Qwen2.5-7b | 0.500000|
> > > | Qwen2.5-7b **I**  | 0.300000|
> > > | Qwen3-8b | 0.900000|
> > > | Qwen3-8b **B**  | 1.000000|
> > >
> > > ## SCITAB Dataset
> > > | Model | Spearman Correlation|
> > > |-------|---------------------|
> > > | Llama2-7b | -0.307794|
> > > | Llama2-7b **C** | 0.820783|
> > > | Llama3-8b | 0.615587|
> > > | Llama3-8b **I**  | 0.111803|
> > > | Mistral-7b | 0.000|
> > > | Mistral-7b **I**  | 0.700000|
> > > | Qwen2.5-7b | 0.335410|
> > > | Qwen2.5-7b **I**  | 0.900000|
> > > | Qwen3-8b | 0.500000|
> > > | Qwen3-8b **B**  | -0.100000|
> > >
> > > Note that the labels **B**, **C**, and **I** refer to the Base, Chat, and Instruct variants of the model.
> > > We find that Llama2-7B shows a striking improvement with the chat tuning, jumping from no correlation (ρ = 0.0) to a robust ρ = 0.7 on WTQ, while also improving on other datasets. Qwen3-8B’s perfect correlation on TAT-QA (ρ = 1.0) suggests highly sensitive and aligned attention behavior, but its negative correlation on SCITAB indicates fragility in the scientific verification task. Overall, the results support the paper’s core finding: attention entropy shifts correlate with performance drops, but this relationship varies mostly by the model's performance.

---

### Decision · Action_Editor_YnB1 · 2025-08-22

**Recommendation:** Accept with minor revision

**Audience:**

Yes

**Audience Explanation:**

The study addresses an important and timely topic in the field of natural language processing, specifically the robustness of large language models in structured data tasks. The findings are relevant to researchers working on improving the reliability and interpretability of language models, as well as those interested in applications involving tabular data.

**Claims And Evidence:**

Yes

**Claims Explanation:**

The authors have provided extensive empirical evidence through a variety of experiments on different datasets (WTQ, TAT-QA, SCITAB) and models (Llama2, Llama3, Mistral, Qwen2.5, Qwen3, TAPEX). They have also included detailed attention analysis and correlation studies, which support their claims regarding the robustness and attention mechanisms of large language models in tabular question answering tasks.